# REVELA: DENSE RETRIEVER LEARNING VIA LANGUAGE MODELING

**Fengyu Cai**[1]    **Tong Chen**[2]    **Xinran Zhao**[3]    **Sihao Chen**[4]    **Hongming Zhang**[5]

**Sherry Tongshuang Wu**[3]    **Iryna Gurevych**[1]    **Heinz Koeppl**[1]

[1]Technical University of Darmstadt    [2]University of Washington

[3]Carnegie Mellon University    [4]Microsoft    [5]Tencent AI Lab

{fengyu.cai, heinz.koeppl}@tu-darmstadt.de

## ABSTRACT

Dense retrievers play a vital role in accessing external and specialized knowledge to augment language models (LMs). Training dense retrievers typically requires annotated query-document pairs, which are costly to create and scarce in specialized domains (e.g., code) or in complex settings (e.g., requiring reasoning). These practical challenges have sparked growing interest in self-supervised retriever learning. Since LMs are trained to capture token-level dependencies through a *self-supervised* learning objective (i.e., next token prediction), we can analogously cast retrieval as learning dependencies among chunks of tokens. This analogy naturally leads to the question: *How can we adapt self-supervised learning objectives in the spirit of language modeling to train retrievers?*

To answer this question, we introduce Revela, a unified and scalable training framework for self-supervised retriever learning via language modeling. Revela models semantic dependencies among documents by conditioning next token prediction on local and cross-document context through an *in-batch attention* mechanism. This attention is weighted by retriever-computed similarity scores, enabling the retriever to be optimized as part of language modeling. We evaluate Revela on domain-specific (CoIR), reasoning-intensive (BRIGHT), and general-domain (BEIR) benchmarks across various retriever backbones. Without annotated or synthetic query-document pairs, Revela surpasses larger supervised models and proprietary APIs on both CoIR and BRIGHT. It achieves BEIR's unsupervised SoTA with ~ 1000x less training data and 10x less compute. Performance increases with batch size and model size, highlighting Revela's scalability and its promise for self-supervised retriever learning.

## 1 INTRODUCTION

Central to information retrieval are dense retrievers (Reimers & Gurevych, 2019; Karpukhin et al., 2020; Ma et al., 2024), which map queries and documents into high-dimensional vector spaces and determine relevance through similarity calculations. Typically, these models rely on carefully annotated query-document pairs and hard negatives for training. However, creating such high-quality training data requires substantial human annotation, which is labor-intensive and difficult to scale in *complex*, *domain-specific* scenarios such as law (Feng et al., 2024) and programming (Jimenez et al., 2024). This limitation stimulates the interest within the community to explore self-supervised approaches for training retrievers directly from unannotated raw texts, i.e., self-supervised retriever learning (Izacard et al., 2022).

Modern LMs (Grattafiori et al., 2024), a successful case of self-supervised learning, are typically pretrained with the next-token prediction (NTP) paradigm, modeling dependencies among

---

 https://github.com/TRUMANCFY/Revela

🤗 https://huggingface.co/trumancai/Revela-3b

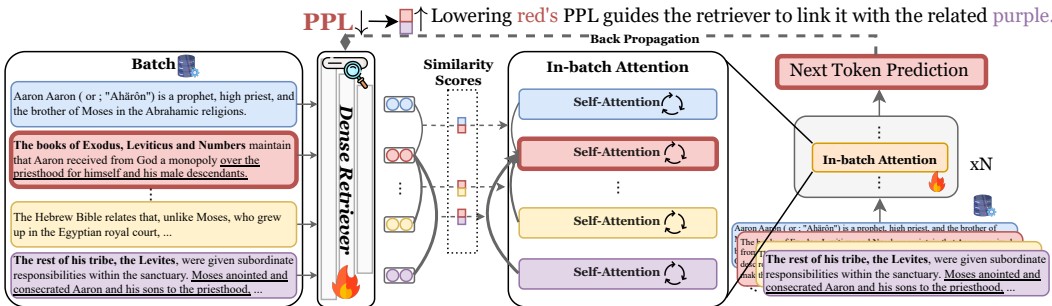

Figure 1: **The framework of Revela.** The retriever's in-batch similarity scores are used as in-batch attention weights inside transformer blocks. The retriever is trained by optimizing the language modeling objective, i.e., NTP. The related patterns in red and purple sequences are highlighted in **bold** and underline. An example of training dynamics is illustrated at App. A.

tokens within a single sequence. Analogously, retriever aims to model relationships among larger units—chunks of tokens—capturing more macroscopic dependencies. This motivates us to raise a key question: *How can we learn a retriever within a self-supervised learning framework of language modeling?* While NTP implicitly identifies the most relevant parts in the context during generation, conditioning language modeling of one sequence on others can serve as an effective proxy for modeling inter-sequence relationships. This can offer a novel and principled approach to self-supervised retriever learning.

In this work, we introduce a self-supervised retriever learning paradigm – Dense Retriever Learning via Language Modeling, abbreviated as Revela. As illustrated in Fig. 1, Revela trains retrievers by simultaneously optimizing retrievers and LMs; different from conventional NTP, Revela learns the probability of a token given both the prefix in this sequence and all other sequences in the batch. Specifically, in addition to classical self-attention which restricts NTP to individual sequences, an **in-batch attention** mechanism enables sequences to attend to their neighbors in the same batch during language modeling. In this process, the retriever provides inter-sequence dependencies that modulate the in-batch attention weights, allowing it to be optimized with the LM during training. We split raw texts into chunks within each document and put these chunks into the same batch, motivated by the idea of hard negative samples in contrastive learning (Xiong et al., 2020).

We comprehensively demonstrate the effectiveness of Revela across three benchmarks in § 4: CoIR (Li et al., 2025), a benchmark tailored for code retrieval, BRIGHT (Hongjin et al., 2025), a reasoning-intensive benchmark across diverse domains, and BEIR (Thakur et al., 2021), a heterogeneous benchmark covering general domains. To do this, we train Revela on Wikipedia for general retrieval and on code-related corpora (Wang et al., 2025) for code retrieval, using pretrained transformers ranging from 135M to 3B parameters as retriever backbones paired with an LM. On CoIR, Revela outperforms a strong, 7B-parameter *supervised* retriever (E5-Mistral-7b-Instruct) by 2.8 points (nDCG@10) and surpasses the weakly-supervised baseline E5-PT (Wang et al., 2022) by 9.7 points at a similar scale. This is particularly noteworthy as both baselines was pre-trained on massive query-document pairs that encompass Revela's training data. Furthermore, Revela outperforms proprietary models on BRIGHT and achieves parity with weakly-supervised E5 model on BEIR while using approximately 1,000x less training data and 10x less compute. These results establish Revela as a highly effective and efficient self-supervised solution. In § 5, we also show that scaling Revela via on larger retriever *backbones*, larger *LMs*, and larger *batchs* can yield greater gain over baselines. Compared to conventional contrastive learning, Revela exhibits stronger cross-domain generalization. Moreover, the *mixed-data* training allows the model to scale across multiple domains while guaranteeing high domain-specific performance. Collectively, the evidence highlights Revela's robust scaling behavior and strong generalization across models, data, and domains. To this end, we summarize our contributions as follows:

- We introduce Revela, a self-supervised framework that trains a retriever via language modeling using an additional in-batch attention mechanism, where next-token prediction is conditioned on both the input sequence and others within the same batch.

- Without query-document pairs, `Revela` surpasses E5-Mistral-7b-Instruct on CoIR by 2.8% (nDCG@10) and outperforms unsupervised baselines by 9.7% at the comparable scale, while also outperforming proprietary APIs on BRIGHT, a challenging, reasoning-intensive benchmark.
- `Revela` exhibits robust scalability across larger models, batch sizes, and mixed-domain data; it also exhibits stronger cross-domain generalization than unsupervised contrastive methods.

## 2  RELATED WORKS

**Self-supervised Retriever Learning**   Dense retrievers are typically trained with query-document pairs, requiring extensive human annotation. Given the abundance of unlabeled corpora, a key challenge in the community is: *How can we train a dense retriever in a self-supervised manner*?

Some methods leverage weak supervision from document corpora. Contriever (Izacard et al., 2022) applies contrastive learning, using passages from the same document as positives and in-batch examples as negatives. E5 (Wang et al., 2022) is trained on a massive dataset of query-document pairs from numerous sources. The primary drawback of this direction is the risk of overfitting to structural biases present in the training data. Other training strategies include distillation from existing retrievers and autoencoding. Distillation is exemplified by LaPraDoR (Xu et al., 2022), which enhances dense retrieval by incorporating signals from BM25 (Robertson et al., 2009). Autoencoding methods, such as RetroMAE (Xiao et al., 2022), learn embeddings via sentence reconstruction. A key drawback of autoencoding is the lack of pairwise supervision, which can cause overfitting to low-level details (Steck, 2020).

Our approach departs from the conventional query-document framework. Drawing inspiration from NTP in LMs, our method adapts the language modeling objective, shifting its focus from predicting adjacent tokens to capturing the inherent associations between entire texts (sequences of tokens).

**LM-guided Retriever Learning**   LM-driven query and document expansion, exemplified by Query2Doc (Wang et al., 2023), can be effective but is often computationally costly due to its need for powerful models. Augmenting LMs with relevant information from external knowledge stores not only improves performance in various NLP tasks but also enhances retriever learning. Atlas (Izacard et al., 2023) utilizes cross-attention scores between retrieved documents and the generated output as signal to train the retriever. However, Atlas uses an encoder-decoder architecture, which diverges from the prevailing trend of decoder-only models and requires costly periodical reindexing. In contrast, our work leverages a standard decoder-only architecture to model relationships between text chunks within and across documents, mitigating the need for reindexing.

With the rise of decoder-only LMs, REPLUG (Shi et al., 2024) enhances retrieval by prepending retrieved documents to the queries, training retrievers to produce query-document similarity aligned with the LM's perplexity. However, the perplexity of *frozen* LMs is often poorly calibrated (Geng et al., 2024), resulting in suboptimal retriever learning. This issue can be optimized in `Revela` where retrievers and LMs are updated jointly during language modeling.

**Domain-specific Retrieval**   In pre-training corpora, domain-specific knowledge is both scarce and rapidly evolving (Grossmann et al., 2023; Wen et al., 2025), making effective retrieval in specialized domains critically challenging. To enhance the adaptability of dense retrievers across domains, researchers have explored continual learning (Sachan et al., 2021; Oguz et al., 2022) and task-aware training (Cheng et al., 2023). However, these approaches still rely on query-document pairs from domain-specific datasets. Another approach seeks to simplify domain-specific retrieval for general-purpose dense retrievers. Cai et al. (2024) propose a divide-and-conquer strategy through a mixed-granularity retrieval framework, significantly enhancing dense retriever performance in scientific domains. Our work demonstrates `Revela`'s domain adaptation capability through language modeling on domain-specific raw texts.

# 3 REVELA: DENSE RETRIEVER LEARNING VIA LANGUAGE MODELING

## 3.1 TRAINING OBJECTIVES

LM training typically includes the maximization of NTP for token sequences. Given a batch of documents $\{D_1, D_2, \ldots, D_B\}$ and an LM parameterized by $\Phi$, classical NTP on the token $x_l^i$ in $D_i = \{x_1^i, \ldots, x_L^i\}$, can be calculated as, where $x_{<l}^i$ denotes the tokens preceding $x_l^i$ in $D_i$

$$P(x_l^i) = P_\Phi(x_l^i \mid x_{<l}^i). \tag{1}$$

As shown in Fig. 2, in Revela, the NTP for sequence $i$ is conditioned not only on its own preceding context $x_{<l}^i$ but also on every other document in the batch, $\{D_j\}_{j\neq i}$. Specifically, we introduce a new attention mechanism in the transformer blocks, i.e., in-batch attention (§ 3.2), which is supported by a specific mask design (§ 3.4). The attentions are weighted by the inter-sequence similarity computed by the retriever parameterized by $\Theta$, i.e., $\mathrm{Sim}(D_i, D_j)$ (§ 3.3), where $\sum_{j\neq i}^B \mathrm{Sim}(D_i, D_j) = 1$. The retriever is dynamically optimized as the similarity is updated jointly with the NTP objective

$$P_R(x_l^i) = P_{\Phi,\Theta}(x_l^i \mid x_{<l}^i, \{D_j\}_{j\neq i}). \tag{2}$$

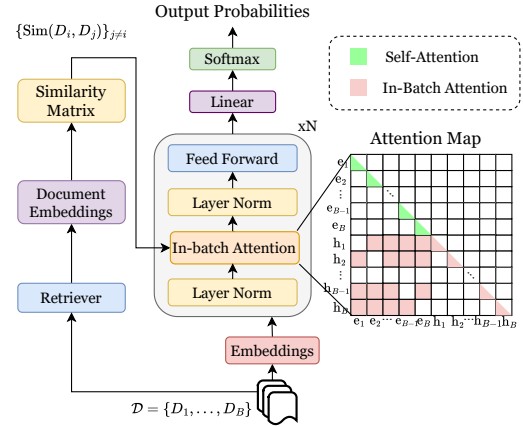

Figure 2: **Revela's architecture.** With an attention map, the embeddings of in-batch attention $\{\mathrm{h}_i^l\}_{i=1}^B$ can attend to the self-attention $\{\mathrm{e}_i^l\}_{i=1}^B$.

## 3.2 IN-BATCH ATTENTION

Revela augments the transformer block with an **in-batch attention** mechanism to incorporate context from other sequences. We denote the output of $D_i$ at the $l$-th layer as $[\mathrm{e}_i^l; \mathrm{h}_i^l]$, where $\mathrm{e}_i^l, \mathrm{h}_i^l \in \mathbb{R}^{L\times d}$ represent the outputs from self-attention and in-batch attention, respectively. For simplicity, we omit layer normalization and feed-forward layers, so the input to the two attention modules in the $l$-th layer is $[\mathrm{e}_i^{l-1}; \mathrm{h}_i^{l-1}]$.

**Standard Self-Attention** In the $l$-th layer of the blocks, the self-attention mechanism computes

$$Q_i^e = \mathrm{e}_i^{l-1}W^Q, \quad K_i^e = \mathrm{e}_i^{l-1}W^K, \quad V_i^e = \mathrm{e}_i^{l-1}W^V, \tag{3}$$

where $W^Q, W^K, W^V \in \mathbb{R}^{d\times d}$ are learnable projection matrices. For multi-head attention with $H$ heads (each of dimension $d_H = d/H$), the standard *causal* attention is computed as

$$\mathrm{e}_i^l = \mathrm{softmax}\left(\frac{Q_i^e K_i^{e\top}}{\sqrt{d_H}}\right) V_i^e. \tag{4}$$

**In-batch attention** combines standard self-attention with cross-document attention. The embeddings of $k$-th token in $D_i$ is obtained by (1) the prefix of $D_i$, and (2) the other documents $\{D_j\}_{j\neq i}$ based on their similarity to $D_i$. This encourages $D_i$ to selectively attend to more relevant documents based on learned retrieval signals.

For the contribution from the prefix of $D_i$, the self-attention output[1] $\mathrm{s}_i^l$ in the in-batch attention uses the same projection matrices as above

$$Q_i^h = \mathrm{h}_i^{l-1}W^Q, \quad K_i^h = \mathrm{h}_i^{l-1}W^K, \quad V_i^h = \mathrm{h}_i^{l-1}W^V, \quad \mathrm{s}_i^l = \mathrm{softmax}\left(\frac{Q_i^h K_i^{h\top}}{\sqrt{d_H}}\right) V_i^h.$$

---

[1]Note that the self-attention output $\mathrm{s}_i^l$ is distinct from the computation of $\mathrm{e}_i^l$. As shown in Fig. 2, $\mathrm{e}_i^l$ and $\mathrm{s}_i^l$ correspond to the upper-left and bottom-right corners of the attention map, respectively. Functionally, $\mathrm{e}_i^l$ captures only sequence-level information, whereas $\mathrm{s}_i^l$ contains in-batch information derived from $\mathrm{h}_i^{l-1}$.

***Cross-document attention*** enables $D_i$ to attend to other documents $D_j$ using cached keys $K_j^e$ and values $V_j^e$. With a *full* attention mask, the bottom-left corner in Fig. 2, the output $\mathrm{b}_{ij}^l$ is computed as

$$\mathrm{b}_{ij}^l = \mathrm{softmax}\left(\frac{Q_i^h K_j^{e\top}}{\sqrt{d_H}}\right) V_j^e. \tag{5}$$

***Weighting by cross-document similarity*** aggregates the attention outputs $\mathrm{b}_{ij}^l$ using the similarity scores $\mathrm{Sim}(D_i, D_j)$ computed by the retriever

$$\mathrm{b}_i^l = \sum_{j=1, j\neq i}^{B} \mathrm{Sim}(D_i, D_j)\, \mathrm{b}_{ij}^l. \tag{6}$$

***Combined output*** integrates the results of self-attention and cross-document attention to form the final output of the in-batch attention

$$\mathrm{h}_i^l = \mathrm{s}_i^l + \mathrm{b}_i^l. \tag{7}$$

### 3.3 Similarity Computation

Given a batch of $B$ documents $\{D_i\}_{i=1}^B$, the retrieval mechanism proceeds in three steps. First, each document $D_i$ is encoded into an embedding $\mathbf{h}_i \in \mathbb{R}^{d_E}$ using an encoder $E_\Theta$, such that $\mathbf{h}_i = E_\Theta(D_i)$. Second, the embeddings are normalized and pairwise cosine similarities are computed: $\tilde{\mathbf{h}}_i = \mathbf{h}_i/\|\mathbf{h}_i\|_2$, and the similarity score between documents $i$ and $j$ is given by $S_{ij} = \tilde{\mathbf{h}}_i^\top \tilde{\mathbf{h}}_j$. Third, temperature-scaled softmax with temperature $\tau$ is applied to obtain probabilities across documents

$$\mathrm{Sim}(D_i, D_j) = \frac{\exp(S_{ij}/\tau)}{\sum_{k\neq i} \exp(S_{ik}/\tau)}.$$

The resulting pairwise weights $[\mathrm{Sim}(D_i, D_j)]_{i,j=1}^B \in \mathbb{R}^{B\times B}$ capture cross-document similarities, allowing the model to condition on relevant in-batch documents.

### 3.4 Implementation Details

As described earlier, `Revela` adapts the classical transformers by additionally including in-batch attention, which builds upon standard self-attention, as shown in Eq. (5). For the minimum modifications to the existing transformer's implementation, we take the computation of $\mathrm{e}$ and $\mathrm{h}$ as duplicating documents and adjusting the attention mask, as illustrated in Fig. 2. In this way, the embeddings $\{\mathrm{h}_i^l\}_{i=1}^B$ produced by in-batch attention can be obtained by applying full attention over the self-attention outputs $\{\mathrm{e}_i^l\}_{i=1}^B$ and aggregating them. `Revela`'s efficiency is discussed in App. C.7.

## 4 Experiments

### 4.1 Experimental Setups

**Evaluation Benchmarks**  To comprehensively evaluate our proposed framework, we benchmark `Revela` on three diverse datasets: CoIR (Li et al., 2025), a comprehensive benchmark designed for *code*-specific retrieval tasks, BRIGHT (Hongjin et al., 2025), a retrieval benchmark requiring intensive *reasoning* to retrieve relevant documents spanning diverse domains, and BEIR (Thakur et al., 2021), a heterogeneous benchmark covering multiple domains for *general* information retrieval. A more detailed introduction is listed in App. B.1.

**Training Data**  One of the earlier baselines for weakly supervised retrievers is E5 (Wang et al., 2022), which collected *1.3B* text pairs from diverse sources such as StackExchange, Wikipedia, Reddit, and scientific papers, and filtered them down to *270M* pairs using handcrafted rules. For `Revela`, We simply convert two E5 pretraining **subsets** to our training corpus: StackOverflow for code-related retrieval (CoIR) and Wikipedia for reasoning-intensive and general retrieval (BRIGHT & BEIR). Data preparation and illustrative examples of these subsets are provided in App. B.2.

- **CoIR**: We segment a set of code-related corpora (Wang et al., 2025) into chunks of at most 120 words, each comprising complete sentences, including the posts in the StackOverflow forum (Weber et al., 2024), online tutorials (Overwijk et al., 2022), and library documentations (Zhou et al., 2023), for CoIR. The batch size is 16. Overall, there are 358,763 training batches.
- **BRIGHT & BEIR**: Similarly, we segment the passages in the Wikipedia corpus.[2] Given a set of passages $\{d_1, d_2, \ldots, d_n\}$, where each passage $d_i$ is divided into chunks $(d_{i1}, d_{i2}, \ldots, d_{im_i})$, the chunks are interleaved in the order $(d_{11}, d_{12}, \ldots, d_{1m_1}, d_{21}, d_{22}, \ldots)$ and then grouped sequentially into batches of size 16. In total, we sample 320,000 batches from 339,409 documents for training. Notably, a *single* batch may contain chunks from *different* documents, highlighting the flexibility of batch construction in Revela.

**Models**  Revela jointly trains a retriever and an LM using the NTP objective. We adopt LLaMA-3.2-1B (Grattafiori et al., 2024) as the LM. To ensure a fair comparison across diverse unsupervised baselines, we adopt a range of LMs with parameter sizes from 0.1B to 3B as the backbone models for the retrievers: SmolLM2-135M (Allal et al., 2025), Qwen2.5-0.5B (Yang et al., 2024), LLaMA-3.2-1B and LLaMA-3.2-3B.[3] We follow the approach used in RepLLaMA (Ma et al., 2024), appending the `<eos>` token to each sentence and use its corresponding embedding as the sentence representation. Additionally, we prepend the prefixes "Query: " and "Passage: " to queries and passages, respectively. For more details about model checkpoints, please refer to App. B.4.

**Baselines**  We include representative self-supervised baselines: **E5-PT**large (E5-PT; Wang et al. 2022) is trained with a contrastive objective, leveraging weak supervision signals from a curated large-scale dataset of text pairs (1.3B raw pairs, filtered to 270M) spanning multiple domains (e.g., code) and covering Revela's training corpus. **REPLUG** (Shi et al., 2024) distills supervision from a frozen LM into a retriever by using LM perplexity to model within-batch chunk–chunk similarity, conditioning one chunk on the other. We adopt REPLUG[4] as our main baseline because (1) it matches Revela's retriever as decoder-only LM design, unlike encoder–decoder systems such as Atlas (Izacard et al., 2023), keeping the focus on joint retriever–LM training; (2) this architectural match makes comparisons generalizable across scales; and (3) REPLUG outperforms most prior methods, making it representative of this line of work (Shi et al., 2024). Consistent with Revela, we use LLaMA-3.2-1B as the frozen reference LM in REPLUG.

For CoIR, we include several supervised retrievers, as well as API-based models such as `OpenAI-Ada-002` (Ada-2) and `Voyage-Code-002` (Voyage-2), following the original setup (Li et al., 2025). Supervised retrievers include UniXcoder (UniX; Guo et al. 2022), which is finetuned on code-related datasets; BGE-M3 (BGE; Chen et al. 2024), a supervised model pretrained and finetuned on text pairs (including code); and E5-Mistral-7B-Instruct (E5-Mistral; Wang et al. 2024).

For BRIGHT, we use several strong baselines: API-based models from `text-embedding-3-large` (OpenAI), `cohere-embed-english-v3.0` (Cohere), and `voyage-large-2-instruct` (VoyageAI), as well as E5-Mistral.

For BEIR, we include **Contriever** (Izacard et al., 2022), a BERT-based retriever trained via contrastive learning on unsupervised pairs, and **LaPraDor** (Xu et al., 2022) uses latent-pair contrastive pre-training on C4 (Raffel et al., 2020) and fuses dense scores with BM25 via lexicon-enhanced dense retrieval.

For more details about the model access and the Huggingface checkpoints, please refer to App. B.4.

**Experimental Details**  For both retrievers and LMs, we apply LoRA with a rank of 256. Training uses the temperature $\tau$ 1e−4, a learning rate of 1e−4, and 100 warmup steps, following the WarmupDecayLR schedule. We train on 4 A100 80GB GPUs with a gradient accumulation step size of 8. Passages are truncated to 160 tokens, and bf16 mixed precision is enabled. We finetune the models for one epoch, namely, there are 10,000 steps on Wikipedia (~ 44 hours) and around 11,000 steps on code-related texts (~ 48 hours). During inference, the max token length of queries and documents is 2048. For more details of the experimental setups, please refer to App. B.5.

---

[2]https://huggingface.co/datasets/Tevatron/wikipedia-nq-corpus

[3]For computational efficiency, we set smaller batch size for the 3B model, e.g., 8.

[4]For replication details, see App. B.3.

Table 1: **Performance on CoIR** (nDCG@10, %). Gray indicates *supervised* models. **Bold** marks the highest score among non-API models in each row. Columns marked $^\dagger$ used code-related pairs during pre-training. The results of APIs are collected from Li et al. (2025). Without query-document pairs, $\text{Revela}_{3B}$ surpasses larger supervised models and proprietary APIs, averaged across 10 tasks.

| Dataset | BM25 | UniX | Revela | E5-PT$^\dagger$ | Revela | BGE$^\dagger$ | REPLUG | Revela | REPLUG | Revela | E5-Mistral$^\dagger$ | Ada-2 | Voyage-2 |
|---|---|---|---|---|---|---|---|---|---|---|---|---|---|
| **Model Size** | – | 0.1B | 0.1B | 0.3B | 0.5B | 0.6B | 1B | 1B | 3B | 3B | 7B | – | – |
| Apps | 4.8 | 1.4 | 8.2 | 10.6 | 20.5 | 14.7 | 13.9 | 19.4 | 17.7 | **26.6** | 23.5 | 8.7 | 26.5 |
| CosQA | 15.6 | 25.1 | 26.2 | 27.1 | 27.5 | 26.4 | 20.1 | 30.2 | 25.2 | 29.0 | **33.2** | 28.9 | 29.8 |
| ST2SQL | 25.0 | 50.5 | 45.7 | 48.9 | 53.7 | 46.9 | 53.9 | 55.0 | 56.8 | 55.9 | **68.0** | 58.3 | 69.3 |
| SN | 40.9 | 60.2 | 49.9 | 35.2 | 57.9 | 58.3 | 50.8 | 64.0 | 53.7 | 62.6 | **67.4** | 74.2 | 81.8 |
| SNCC | 54.0 | 58.4 | 63.4 | 50.5 | 68.0 | 53.7 | 62.8 | **70.0** | 62.8 | 69.1 | 64.8 | 69.1 | 73.5 |
| TransC | 47.8 | 41.8 | 70.9 | 56.3 | 77.6 | 62.6 | 61.5 | 81.1 | 62.1 | **83.2** | 80.6 | 53.3 | 72.8 |
| TransDL | 34.4 | 31.0 | 34.6 | 32.2 | **35.4** | 30.2 | 33.3 | 34.2 | 34.4 | 34.5 | 31.7 | 26.0 | 27.3 |
| SOQA | 70.2 | 44.7 | 69.2 | 86.9 | 82.5 | 80.6 | 76.3 | 85.7 | 78.1 | 88.3 | **91.0** | 72.4 | 87.7 |
| F-ST | 68.1 | 36.0 | 63.8 | 70.4 | 74.5 | 69.3 | 66.0 | 76.2 | 71.7 | **78.8** | 76.4 | 47.1 | 65.4 |
| F-MT | 59.2 | 24.2 | 51.7 | 46.2 | 63.6 | 47.9 | 42.9 | 70.4 | 49.1 | **73.0** | 36.4 | 17.7 | 28.7 |
| **Mean** | 42.0 | 37.3 | 48.4 | 46.4 | 56.1 | 49.1 | 48.2 | 58.6 | 53.9 | **60.1** | 57.3 | 45.6 | 56.3 |

## 4.2 EXPERIMENTAL RESULTS

**Revela exhibits superior performance on domain-specific retrieval.** Tab. 1 reports the performance of Revela and baseline methods on CoIR. As an unsupervised model trained without query-document pairs, $\text{Revela}_{3B}$ surpasses E5-Mistral-7B-Instruct, a much larger supervised model pre-trained and fine-tuned on massive, well-curated query-doc pairs, as well as two proprietary APIs averaged on 10 tasks. Revela also follows *scaling laws*: its performance consistently improves with larger model sizes, while maintaining superiority at every scale over baselines. At 0.1B parameters, Revela outperforms the code-specific supervised model UniXCoder by 11.1 points on nDCG@10. At the 0.5B scale, our model outperforms E5-PT by nearly 10 points, despite E5-PT being pre-trained on 270 million filtered query-document pairs covering Revela's corpus. $\text{Revela}_{0.5B}$ even surpasses the supervised model BGE-M3, despite the latter being pre-trained on extensive text-code pairs.[5] Moreover, at each scale, Revela also surpasses REPLUG, underscoring the effectiveness of the retriever-LM *co-training* paradigm for retriever learning.

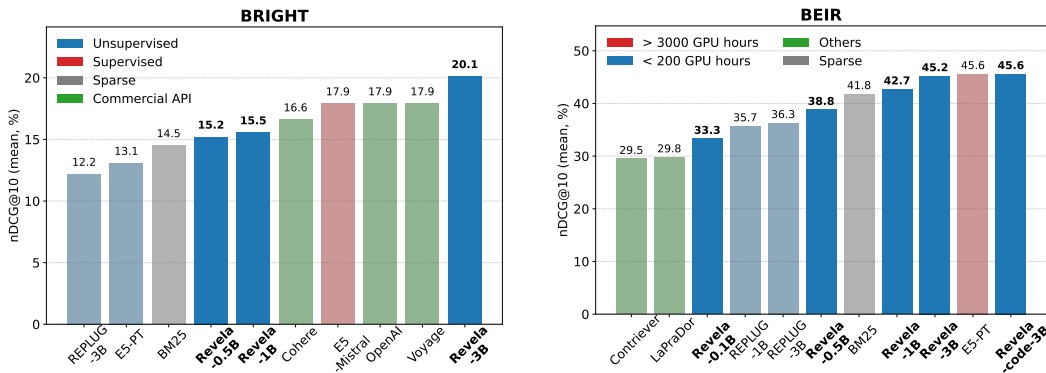

Figure 3: **Performance on BRIGHT (left) and BEIR (right)** (nDCG@10, %). Results for Revela are shown in opaque bars, while all other models are represented by transparent bars. On BRIGHT, $\text{Revela}_{3B}$ surpasses E5-Mistral, a supervised retriever with more parameters, and properties APIs. On BEIR, Revela achieves similar performance with E5-PT with much less data and compute. Please refer to Tab. 7 and Tab. 8 in App. B.6 for the per-task results.

**Revela demonstrates strong performance on complex retrieval.** Fig. 3 (left) shows the average nDCG@10 across the 12 BRIGHT subtasks. Despite being trained unsupervised on only 340K raw Wikipedia documents, $\text{Revela}_{3B}$ outperforms the supervised E5-Mistral-7B-Instruct as

---

[5]Please refer to Table 8 in the original paper (Chen et al., 2024).

well as proprietary embedding models. The scaling trends observed in code retrieval persist here as well: even $\text{Revela}_{0.5B}$ outperforms $\text{REPLUG}_{3B}$ with much fewer parameters. At comparable scale, $\text{Revela}_{0.5B}$ exceeds E5-PT by 3.1 points (23.7% relative), a noteworthy result given the potential advantage from E5-PT's training corpus overlapping with BRIGHT. These findings highlight Revela's promise for tackling more complex retrieval scenarios.

**Revela achieves efficient and robust generalization across tasks.** Fig. 3 (right) reports the average nDCG@10 across the 13 BEIR tasks, where Revela demonstrates remarkable efficiency and robustness. At the 0.1B scale, Revela outperforms Contriever and LaPraDor by over 3 absolute points. Moreover, Revela's consistently surpass REPLUG by a significant margin (3B: 8.9%; 1B: 7.0%), mirroring trends observed on CoIR and BRIGHT. Remarkably, despite using approximately **1000×** less training data and **10×** fewer compute resources, $\text{Revela}_{3B}$ matches E5-PT's performance, underscoring its efficiency. Most notably, when trained solely on a code-related corpus, $\text{Revela}_{3B}$ performs comparably on the general-domain BEIR benchmark to both its Wikipedia-trained counterpart and E5-PT, demonstrating strong cross-domain generalization.

## 5 ANALYSIS

We conduct several targeted analyses to further investigate Revela. First, to isolate its algorithmic contribution, we compare it with Contriever using an identical LM backbone. This comparison reveals that Revela achieves superior performance and exhibits stronger domain-specific robustness to training data. Second, we demonstrate that, consistent with traditional contrastive learning, Revela benefits from larger batch sizes. Finally, we examine the LM's impact on retriever performance within the co-training framework. Additional studies on mixed-domain training, out-of-domain generalization, the LM's post-Revela capabilities, and computational efficiency are presented in App. C.

**Controlled experiments under the same LM backbone.** As baseline models may use different sizes and architectures of LMs as retriever backbones, we implement a classical unsupervised retriever learning algorithm, Contriever (Izacard et al., 2022), using the *same* model (LlaMA-3.2-1B) and the *same* datasets (Wikipedia and the code-related corpus introduced in § 4.1) as training data. To construct pseudo query-document pairs from raw text, Contriever mainly applies two tricks: *Inverse Cloze Task*, where a sentence is removed from a passage to form a query against the remainder, and *Independent Cropping*, where two spans from the same document form a positive pair while spans from different documents serve as negatives. In this way, we generate 500K and 359K query-document pairs, each with 15 negatives, for general and code domains, respectively. The models contrastively trained on them are denoted as $\text{Contriever-wiki}_{1B}$ and $\text{Contriever-code}_{1B}$. Please refer to App. C.1 for more training details.

As shown in Tab. 2, Revela outperforms Contriever on both BEIR and CoIR, irrespective of whether training is conducted on general or code-specific data. Moreover, the performance disparity becomes more pronounced in out-of-distribution domains, underscoring Revela's robustness and its strong capacity for cross-domain generalization over contrastive learning.

Table 2: Revela vs. Contriever Performance.

| Model | BEIR | CoIR | AVG |
|---|---|---|---|
| $\text{Revela-wiki}_{1B}$ | **42.7** | **53.2** | **48.0** |
| $\text{Contriever-wiki}_{1B}$ | 42.4 | 50.3 | 46.4 |
| $\text{Revela-code}_{1B}$ | **39.6** | **58.6** | **49.1** |
| $\text{Contriever-code}_{1B}$ | 32.3 | 52.1 | 42.2 |

**Revela benefits from a larger batch size.** The in-batch attention mechanism in Revela is inspired by the concept of in-batch negatives in supervised contrastive learning (Xiong et al., 2020). As described in § 4.1, the default batch size is set to 16. To analyze the impact of batch size, we construct training data with batch sizes of 4 and 8 using the same batch construction strategy. As shown in Fig. 4, Revela's performance scales with batch size, suggesting potential for further gains. For more detailed results within CoIR and BEIR, please refer to App. C.2.

**Larger LMs can help out-of-domain retrieval tasks.** As observed in the previous section, retriever performance improves with larger backbone models. In this section, we further investigate how the size of the LM, the other key component in training, affects the retriever's effectiveness. In

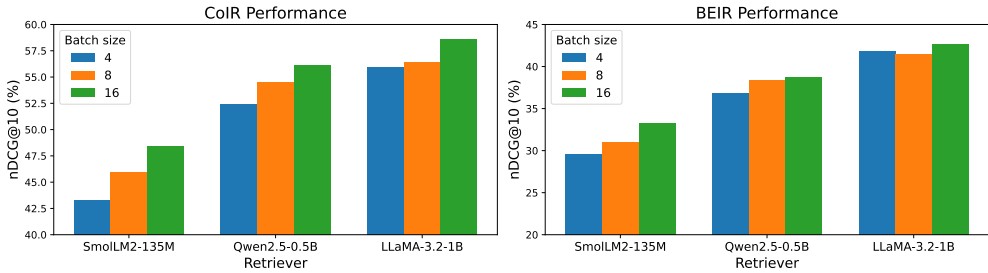

Figure 4: Performance comparison on CoIR and BEIR with different ***batch sizes***. For both benchmarks, `Revela` performance generally scales with batch size.

addition to LLaMA-3.2-1B, we also use SmolLM2-135M and Qwen2.5-0.5B as LMs, each paired with retrievers of three different sizes. All models are trained using the same experimental setup described in § 4, on both training corpora.

As shown in Fig. 5, CoIR exhibits a clear positive trend: the largest LM delivers best retrieval performance. In contrast, on the general-domain BEIR benchmark, larger LMs do not provide a consistent advantage. These findings suggest that incorporating larger LMs will likely enhance `Revela`'s performance on specialized domains while maintaining competitive results on general-domain tasks. For per-dataset results, see App. C.3.

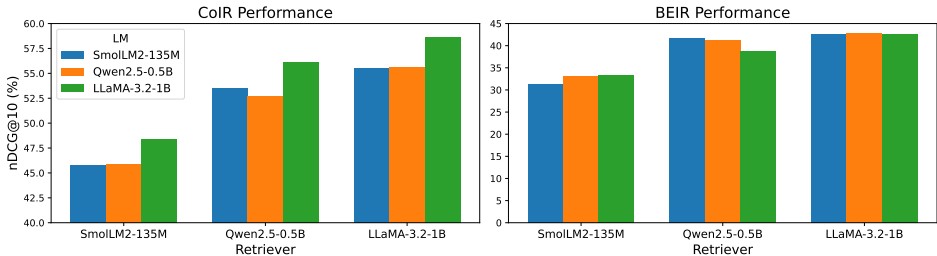

Figure 5: Performance comparison on CoIR and BEIR using various combinations of retrievers and *LMs*. For code retrieval tasks, larger LMs can yield greater gains in retriever performance.

To further investigate `Revela`, we conducted extra experiments, with the following key findings:

- `Revela` learns efficiently from *mixed-domain* training corpora. Mixing Wikipedia with the code-related corpora used in § 4 maintains, or even improves, retrieval performance, demonstrating `Revela`'s strong generalization across *diverse* domains with *only* raw texts (See App. C.4).
- Even when trained on an LM-training, out-of-domain corpus, `Revela` still performs competitively, underscoring its robustness and confirming the observation in Tab. 2. At a similar scale, `Revela`, trained on out-of-domain data, can still outperform E5-PT on CoIR (See App. C.5).
- The co-trained LM's capacity is largely preserved, plausibly due to the use of LoRA and retaining the NTP objective while adding only auxiliary in-batch attention (See App. C.6).
- `Revela` offers a theoretical advantage in *efficiency* over REPLUG, complementing its empirically demonstrated performance gains (See App. C.7).

## 6 FUTURE DIRECTIONS

We train retrievers directly from raw text via self-supervised language modeling, sidestepping query-doc pairs, a breakthrough beyond the traditional paradigm. Based on this novel solution, We outline several future directions to suggest follow-up research. (1) **Iterative Indexing**: While `Revela` uses document chunking, a more general approach would iteratively index documents and group chunks by on-the-fly representations. Though explored in prior work (Izacard et al., 2023; Shi et al., 2024), the high computational cost of such methods remains a key challenge for future work. (2) **Scal-**

**ing up**: We envision several directions to scale up `Revela`, including increasing retriever size in § 4.2, and increasing LM size in § 5. Additionally, incorporating more advanced attention mechanisms (Yuan et al., 2025) may enhance retriever learning and accelerate the training. (3) **Multi-modality**: Although `Revela` targets text and code retrieval, this retriever-via-language-modeling paradigm can, in principle, generalize to modalities, such as images (Jiang et al., 2025).

## 7 CONCLUSION

Efficiently building information-seeking systems is crucial due to the swiftly evolving world and the wide existence of specific domains, where query-document curation is one key bottleneck. In this work, we introduce `Revela`, a self-supervised framework that couples dense retrieval with language modeling through a novel in-batch attention mechanism, where a token attends to local context and other sequences in the batch during NTP. By letting the retriever's relevance scores weight cross-sequence attention, `Revela` transforms NTP into a retrieval signal, making use of *raw text* and eliminating the need for annotation or synthesis. Our experiments on domain-specific, complex, and general benchmarks demonstrate significant performance gains over existing self-supervised methods, with improvements scaling with retriever size. Further analysis on batch size, LM size, and mixed-data composition highlights `Revela` as a strong and scalable alternative to traditional self-supervised paradigms, paving the way for more general and efficient retriever learning.

## REPRODUCIBILITY STATEMENT

To ensure full reproducibility, we provide comprehensive details on our methods and experiments. The setups for our main results, presented in § 4.2, are described in § 4.1, including hyperparameters, training resources, training time, etc. Further implementation details are located in the appendices, covering our model, `Revela` (App. B.5), REPLUG implementation (App. B.3) with its checkpoints (App. B.4), evaluation benchmarks (App. B.1), and the training corpus (App. B.2).

The appendices also contain our extended analyses. These include a detailed comparison with unsupervised contrastive learning methods (App. C.1), a study on mixed-domain composition within the training data (App. C.4), and an outline of data usage for out-of-domain generalization (App. C.5). The code to reproduce our results is included in the supplementary submission.

## ACKNOWLEDGMENTS

We thank Kexin Wang, Sheng Lu, Jan Buchmann, Falko Helm, and Yuhao Zhang for the discussion and comments on an early draft of this work.

Fengyu Cai is funded by the German Federal Ministry of Education and Research and the Hessian Ministry of Higher Education, Research, Science, and the Arts within their joint support of the National Research Center for Applied Cybersecurity ATHENE, and by the Federal Ministry of Education and Research as part of the Software Campus 3.0 project ETRAG (funding code 16IS23067). This work is also supported by the Distr@l4a funding line of the State of Hesse (project number: 493 24_0015_4A). Xinran Zhao is supported by the ONR Award N000142312840.

This work has been supported by the European Union (ERC, InterText, 101054961). Views and opinions expressed are however those of the author(s) only and do not necessarily reflect those of the European Union or the European Research Council. Neither the European Union nor the granting authority can be held responsible for them. The work is related to WP1 on cross-document modelling.

We gratefully acknowledge support from the hessian.AI Service Center (funded by the Federal Ministry of Research, Technology and Space, BMFTR, grant no. 16IS22091) and the hessian.AI Innovation Lab (funded by the Hessian Ministry for Digital Strategy and Innovation, grant no. S-DIW04/0013/003).

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

# A ILLUSTRATION

To illustrate the training dynamics, we visualize both the in-batch attention computed by the retriever and the LM loss, for example, in Fig. 6. We use LLaMA-3.2-1B for both the retriever and the LM, and compare at the initial checkpoint, 100 steps, and 200 steps. When the retrievers can model the semantics, the NTP loss decreases as shown in Fig. 6. In this example, (Blue, Yellow) and (Red, Purple) are semantically relevant pairs, where the former is related to biographical information of Aaron, while the latter is related to Aaron's priesthood and religious duties

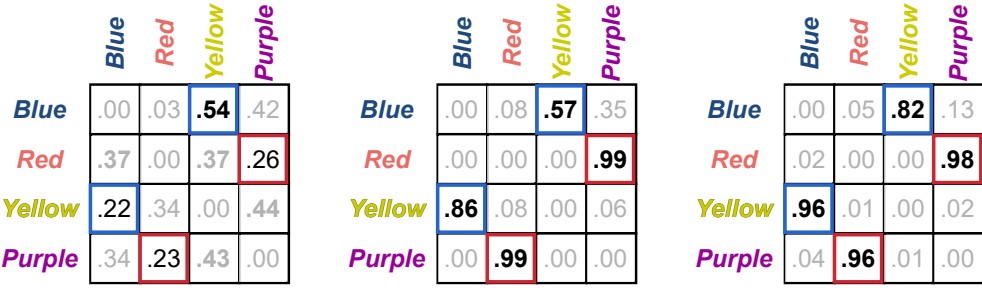

Figure 6: Example of training dynamics of Revela. The related patterns in red and purple sequences are highlighted in **bold**, underline, and *italic*.

# B EXPERIMENTS

## B.1 EVALUATION BENCHMARKS

**CoIR** contains four classes of code-related retrieval tasks, including Text-to-Code Retrieval[a], Code-to-Code Retrieval[b], Code-to-Text Retrieval[c], and Hybrid Code Retrieval[d]. Table 3 presents the specific subtasks included in CoIR with the corresponding descriptions.

Table 3: **CoIR Benchmark Tasks.** The superscripts present the type of the tasks. The abbreviation of the tasks is noted in the parentheses, presented in Table 1.

| Sub-Tasks (Abbr.) | Descriptions |
|---|---|
| AppsRetrieval (Apps)[a] | Retrieve code snippets based on natural language queries. |
| CosQA (CosQA)[a] | Find code snippets that answer web search queries. |
| SyntheticText2SQL (ST2SQL)[a] | Retrieve SQL queries based on natural language questions. |
| COIRCodeSearchNetRetrieval (SN)[b] | Retrieve explanations or summaries for code snippets. |
| CodeSearchNetCCRetrieval (SNCC)[c] | Identify code snippets that are similar to a given one. |
| CodeTransOceanContest (TransC)[c] | Retrieve code solutions based on contest problems. |
| CodeTransOceanDL (TransDL)[c] | Retrieve relevant deep learning code contexts or modules. |
| StackOverflowQA (SOQA)[d] | Handle hybrid code-related QA tasks with both text and code. |
| CodeFeedbackST (F-ST)[d] | Retrieve answers to single-turn code-related questions. |
| CodeFeedbackMT (F-MT)[d] | Handle multi-turn question-answer scenarios in code. |

**BRIGHT** is a benchmark for reasoning-intensive retrieval across diverse domains, including Stack-Exchange forums, coding tasks, and theorem-based math. It emphasizes cases with little lexical overlap between queries and relevant documents, exposing the limitations of existing models. Table 4 presents the specific subtaks included in BRIGHT with the corresponding descriptions.

Table 4: **BRIGHT Benchmark Tasks.** Abbreviations and descriptions.

| Abbrev. | Description |
|---|---|
| Bio. | StackExchange Biology questions with external web evidence |
| Earth. | StackExchange Earth Science questions with linked resources |
| Econ. | StackExchange Economics questions requiring theoretical support |
| Psy. | StackExchange Psychology questions with cited references |
| Rob. | StackExchange Robotics questions involving technical docs |
| Stack. | StackOverflow programming questions with linked web pages |
| Sus. | StackExchange Sustainable Living questions with supporting docs |
| Leet. | LeetCode problems; retrieve similar problems/solutions |
| Pony. | Pony language coding tasks with syntax documentation |
| AoPS | Math Olympiad problems; retrieve others using same skill |
| TheoQ. | TheoremQA: retrieve problems applying the same theorem |
| TheoT. | TheoremQA: retrieve theorems from ProofWiki relevant to query |

## B.2 Examples of the Training Corpus

Fig. 7 presents an example batch containing 16 chunks from two topics. The sentences are chunked by NLTK,[6] while the maximum length of a chunk is limited to 120 words. When training, the chunks will be randomly shuffled within the batch.

## B.3 REPLUG

**Self-supervised batch loss.** In our setup, as there are no external queries, each document plays the role of both query and context. For a batch $\mathcal{D} = \{D_1, \ldots, D_B\}$, we embed every document with the retriever $E_\Theta$:

$$s_{ij} = \frac{E_\Theta(D_i)^\top E_\Theta(D_j)}{\tau_r}, \qquad P_\Theta(j \mid i, \mathcal{D}) = \frac{\exp(s_{ij})}{\sum_{k=1}^{B} \exp(s_{ik})}.$$

A frozen LM $g_\Phi$ measures how well $D_j$ explains $D_i$:

$$\ell_{ij} = -\log g_\Phi(D_i \mid D_j), \qquad P_\Phi(j \mid i, \mathcal{D}) = \frac{\exp(-\ell_{ij}/\tau_m)}{\sum_{k=1}^{B} \exp(-\ell_{ik}/\tau_m)}.$$

We train the retriever by aligning these two distributions for every anchor document:

---

[6]https://www.nltk.org/

---

**Alcohol – chemistry, properties, and reactions**

1. *C–C-bond formation and reductions: Barbier, Nozaki–Hiyama, $NaBH_4/LiAlH_4$, Meerwein–Ponndorf–Verley, and Noyori asymmetric hydrogenation.*
2. *Alcohols have $pK_a$ 16–19; deprotonation by strong bases yields alkoxides, yet neutral OH is a poor leaving group.*
3. *Protonation ($R–OH \rightarrow R–OH_2{}^+$) activates SN1/SN2 substitution; e.g. tertiary alcohol + HCl tert-alkyl chloride, or thionyl chloride for primary/secondary cases.*
4. *Hydrobromic acid or $PBr_3$ give alkyl bromides; Barton–McCombie deoxygenates alcohols ...*
5. *E1 elimination: acid-catalysed dehydration follows Zaitsev, fastest for tertiary alcohols; Fischer esterification and tosylation convert OH into esters.*
6. *Oxidations: primary aldehyde/carboxylic acid, secondary ketone, tertiary inert; pathway proceeds via hydrate ($R–CH(OH)_2$).*
7. *Typical oxidants: Collins reagent, Dess–Martin periodinane, $KMnO_4$, Jones reagent; alcohol = any molecule with a C–OH group.*

**Achill Island – geography, history, and culture**

8. *Largest Irish island (area 148 $km^2$, pop. 2700); linked to the mainland by Michael Davitt Bridge*
9. *First settlers circa 3000 BC; 87 % peat bog; late-Neolithic population estimate 500–1 000.*
10. *Deforestation for cultivation; Iron-Age promontory forts at Slievemore, Atlantic Drive, Achillbeg testify to a martial past.*
11. *Umhall territory ruled by the seafaring O'Malleys; Butler/de Burgo control after Anglo-Norman invasion; 17th–18th-century inward migration.*
12. *Two Irish dialects co-existed; Carrickkildavnet Castle (15th c.) and the exploits of Grace O'Malley embody maritime heritage.*
13. *Grace O'Malley met Elizabeth I in 1593; Rev. Edward Nangle founded the proselytising Achill Mission ("the Colony") in 1831.*
14. *Mission thrived then waned; Westport–Newport railway ..., echoing Brian Rua O'Cearbhain's prophecy of "carts on iron wheels".*
15. *First train carried victims of the Clew Bay drowning (1894); ..neared with the Kirkintilloch bothy fire (1937).*
16. *Rail line closed weeks after 1937 ... St Dymphna's holy well stand on the south-east coast.*

Figure 7: Example of one batch containing chunks split from Wikipedia: This batch contains chunks from two topics: Alcohol (blue) and Achill Island topic (green).

$$\mathcal{L}(\Theta) = \frac{1}{B} \sum_{i=1}^{B} \mathrm{KL}\big(P_\Phi(\cdot \mid i, \mathcal{D}) \parallel P_\Theta(\cdot \mid i, \mathcal{D})\big).$$

LM parameters $\Phi$ remain fixed; only $\Theta$ is optimized.

In our experiments, both of temperatures, $\eta_m$ and $\tau_r$, are set as 0.001. REPLUG is training on the same datasets with `Revela`, with the learning rate $5e^{-4}$ and the training steps 4500. All other experimental setups are identical to `Revela`.

## B.4 CHECKPOINTS

We include the off-the-shelf unsupervised and supervised retrievers and the corresponding hugging-face URLs in Tab. 5. For the prosperity APIs, we list the URLs in Tab. 6.

Table 5: Baseline retrievers, LMs (`Revela`'s backbone), CodeRAG-Bench datasets, and evaluation benchmarks with their HuggingFace URLs and licenses.

| Name | URL | License |
|---|---|---|
| UniXcoder (UniX) | microsoft/unixcoder-base | Apache-2.0 |
| Contriever | facebook/contriever | Not specified |
| RetroMAE | Shitao/RetroMAE | Not specified |
| GraphCodeBERT | microsoft/graphcodebert-base | Not specified |
| LaPraDoR | canwenxu/laprador | Apache-2.0 |
| E5-large-unsupervised (E5-PT$_{large}$) | intfloat/e5-large-unsupervised | MIT |
| BGE-m3 | BAAI/bge-m3 | MIT |
| E5-Mistral-7B-Instruct | intfloat/e5-mistral-7b-instruct | MIT |
| SmolLM2-135M | HuggingFaceTB/SmolLM2-135M | Apache-2.0 |
| Qwen 2.5-0.5B | Qwen/Qwen2.5-0.5B | Apache-2.0 |
| LLaMA-3.2-1B | meta-llama/Llama-3.2-1B | Llama 3.2 Community License Agreement |
| LLaMA-3.2-3B | meta-llama/Llama-3.2-3B | Llama 3.2 Community License Agreement |
| LLaMA-3.1-8B | meta-llama/Llama-3.1-8B | Llama 3.1 Community License Agreement |
| Library Documentation | code-rag-bench/library-documentation | CC BY-SA 4.0 |
| Online Tutorials | code-rag-bench/online-tutorials | CC BY-SA 4.0 |
| StackOverflow Posts | code-rag-bench/stackoverflow-posts | CC BY-SA 4.0 |
| BEIR | BeIR/beir | CC BY-SA 4.0 |
| CoIR | CoIR-Retrieval | Not specified |

Table 6: Embedding models and their reference URLs.

| Name | URL |
|---|---|
| OpenAI-Ada-002 | platform.openai.com/docs/guides/embeddings |
| Voyage-code-2 | blog.voyageai.com/2024/01/23/voyage-code-2-elevate-your-code-retrieval/ |
| text-embedding-3-large | openai.com/index/new-embedding-models-and-api-updates/ |
| voyage-large-2-instruct | docs.voyageai.com/docs/embeddings |
| cohere-embed-english-v3.0 | huggingface.co/Cohere/Cohere-embed-english-light-v3.0 |

## B.5 EXPERIMENTAL SETUPS

**Model Architecture** When trained on Wikipedia, `Revela` prevents any single token from dominating the attention by scaling the output $b_{ij}^l$ with the norm of $V_j^e$, thereby encouraging the cross-document attention to focus on sequence-level semantics (Izacard et al., 2023). This operation, referred to as *V-normalization*, is computed as

$$\tilde{b}_{ij}^l = \frac{b_{ij}^l}{N_{ij} + \epsilon}, \quad \text{where } N_{ij} = \text{softmax}\left(\frac{Q_i^h K_j^{e\top}}{\sqrt{d_H}}\right) \|V_j^e\|_2. \tag{8}$$

$$b_i^l = \sum_{j=1, j\neq i}^{B} \text{Sim}(D_i, D_j)\, \tilde{b}_{ij}^l. \tag{9}$$

where $\epsilon$ is a small constant for numerical stability, set as $1\text{e}-6$ in our experiment. When trained on code-related corpus and evaluated on CoIR, `Revela` performs better without V-normalization. We will leave the exploration of more variants of the architecture design in the future.

**Similarity Calculation within Chunks** When calculating the similarity between chunks derived from Wikipedia, we follow REPLUG, using only the first half of the chunk to compute similarity with other chunks. This design is motivated by the inferential semantics inherent in natural language retrieval tasks. In contrast, for code retrieval tasks, we retain the full chunk when computing similarity, as these tasks rely more heavily on precise semantic matching compared to natural language.

Table 7: Performance of unsupervised/self-supervised retriever models on BEIR datasets (nDCG@10, %). **Bold** marks the best score per dataset among unsupervised methods.

| Dataset | BM25 | Contriever | LaPraDor | Revela | E5 | REPLUG | Revela | REPLUG | Revela | REPLUG | Revela | Revela-code |
|---|---|---|---|---|---|---|---|---|---|---|---|---|
| Model Size | – | 0.1B | 0.1B | 0.1B | 0.3B | 0.5B | 0.5B | 1B | 1B | 3B | 3B | 3B |
| ArguAna | **48.7** | 44.3 | 44.6 | 39.0 | 44.4 | 36.4 | 41.1 | 39.4 | 44.6 | 38.0 | 45.3 | 44.3 |
| ClimateFEVER | 13.6 | 7.2 | 12.2 | 15.3 | 15.7 | 16.2 | 13.8 | 15.0 | 15.8 | 13.0 | 16.6 | **18.6** |
| DBPedia | 29.9 | 27.0 | 25.0 | 18.3 | **37.1** | 18.0 | 21.3 | 19.5 | 27.6 | 23.4 | 27.1 | 30.9 |
| FEVER | 48.1 | 27.2 | 33.6 | 33.7 | **68.6** | 50.4 | 51.1 | 51.6 | 61.7 | 54.3 | 62.9 | 66.3 |
| FiQA2018 | 25.1 | 12.4 | 19.8 | 19.2 | **43.2** | 19.9 | 27.2 | 22.7 | 30.4 | 24.8 | 34.3 | 33.2 |
| HotpotQA | 56.9 | 41.0 | 30.4 | 38.5 | 52.2 | 35.6 | 50.6 | 42.4 | 56.0 | 33.9 | **58.8** | 57.8 |
| NFCorpus | 32.1 | 27.1 | 30.4 | 23.6 | **33.7** | 26.9 | 26.8 | 26.7 | 27.2 | 26.9 | 33.0 | 32.9 |
| NQ | 28.5 | 18.1 | 18.0 | 21.2 | **41.7** | 26.7 | 29.8 | 27.8 | 33.9 | 38.1 | 40.8 | 41.0 |
| QuoraRetrieval | 80.4 | 83.4 | 78.7 | 81.0 | 81.0 | 78.4 | 83.2 | 82.4 | 83.5 | 83.3 | 83.8 | **85.6** |
| SCIDOCS | 15.8 | 10.9 | 13.4 | 12.1 | **21.8** | 13.6 | 14.8 | 14.5 | 16.3 | 15.3 | 17.6 | 18.8 |
| SciFact | 68.7 | 59.1 | 49.9 | 57.9 | 72.3 | 65.0 | 66.0 | 67.6 | 71.9 | **73.7** | 73.3 | 72.0 |
| TRECCOVID | 62.2 | 18.2 | 22.9 | 60.6 | 61.8 | 44.9 | 58.4 | 39.4 | 60.1 | 46.2 | **66.7** | 60.7 |
| Touche2020 | **33.1** | 7.2 | 8.9 | 12.4 | 19.8 | 11.8 | 19.9 | 14.7 | 25.7 | 1.5 | 26.9 | 30.5 |
| Mean | 41.8 | 29.5 | 29.8 | 33.3 | **45.6** | 34.2 | 38.8 | 35.7 | 42.7 | 36.3 | 45.2 | **45.6** |

Table 8: **Performance on BRIGHT** (nDCG@10, %). **Bold** marks the best performance. The results of BM25, E5-Mistral and APIs are taken from BRIGHT (Hongjin et al., 2025).

| Dataset | BM25 | E5-PT | Revela | Revela | REPLUG | Revela | E5-Mistral | Cohere | OpenAI | Voyage |
|---|---|---|---|---|---|---|---|---|---|---|
| Model Size | – | 0.3B | 0.5B | 1B | 3B | 3B | 7B | – | – | – |
| Biology | 18.9 | 18.5 | 16.9 | 16.7 | 6.8 | **24.9** | 18.6 | 18.7 | 23.3 | 23.1 |
| Earth Science | 27.2 | 29.5 | 29.0 | 29.4 | 13.0 | **40.3** | 26.0 | 28.4 | 26.7 | 25.4 |
| Economics | 14.9 | 10.2 | 13.9 | 13.2 | 12.9 | 17.2 | 15.5 | **20.4** | 19.5 | 19.9 |
| Psychology | 12.5 | 17.6 | 13.3 | 15.0 | 15.1 | 21.4 | 15.8 | 21.6 | **27.6** | 24.9 |
| Robotics | 13.6 | 7.1 | 9.1 | 10.5 | 7.4 | 15.3 | **16.3** | **16.3** | 12.8 | 10.8 |
| StackOverflow | **18.4** | 8.7 | 10.8 | 15.5 | 7.5 | 16.9 | 11.2 | 18.3 | 14.3 | 16.8 |
| Sustainable | 15.0 | 14.2 | 13.9 | 15.4 | 12.1 | 19.2 | 18.1 | 17.6 | **20.5** | 15.4 |
| Pony | **7.9** | 1.8 | 3.9 | 3.0 | 1.1 | 6.5 | 4.9 | 1.9 | 2.4 | 1.5 |
| LeetCode | 24.4 | 22.5 | 27.6 | 26.1 | 24.8 | 26.6 | 28.7 | 26.8 | 23.6 | **30.6** |
| AoPS | 6.2 | 3.3 | 12.5 | 8.8 | **13.3** | 11.6 | 7.1 | 6.3 | 8.5 | 7.5 |
| ThmQA-Thm | 4.9 | 6.0 | 6.7 | 8.9 | 9.2 | 14.2 | **26.8** | 7.2 | 11.7 | 11.6 |
| ThmQA-Q | 10.4 | 17.4 | 24.8 | 24.1 | 22.9 | 27.1 | 26.1 | 15.7 | 23.5 | **27.4** |
| Mean | 14.5 | 13.1 | 15.2 | 15.6 | 12.2 | **20.1** | 17.9 | 16.6 | 17.9 | 17.9 |

## B.6 SUPPLEMENTARY RESULTS

Tab. 7 and Tab. 8 present the per-task results in BEIR and BRIGHT, including Revela and other baselines.

# C ANALYSIS

## C.1 EXPERIMENTAL SETUPS OF CONTRIEVER TRAINING

Our Contriever implementation fine-tunes a LoRA-adapted LLaMA-3.2-1B retriever on a 500k Wikipedia and 359k code dataset, using contrastive learning with a temperature of 0.01, EOS pooling, and query/passage prefixes as "Query: " and "Passage: ", respectively. The LoRA rank is 256, consistent with Revela. The training is optimized with DeepSpeed ZeRO-3, bfloat16 precision, gradient checkpointing, and an effective batch size of 256. The number of negatives for each query is 15, which is consistent with the popular contrastive learning setup such as MsMarco.[7]

The per-task results in Tab. 2 on CoIR and BEIR are displayed in Tab. 9 and Tab. 10. For in-domain evaluation, Revela outperforms the contrastive-learning counterpart on both BEIR and CoIR. Though the performance on BEIR is quite close, Revela demonstrate better generalization: it outperforms Contriever on tasks like FiQA, SciFact, and TRECCOVID, while Contriever exhibits better performance on ClimateFEVER or FEVER, which use Wikipedia as the corpus. For out-of-

---

[7]https://huggingface.co/datasets/Tevatron/msmarco-passage-aug

Table 9: Performance on CoIR (nDCG@10, %). **Bold** marks the best score among the models.

| Dataset | Revela-wiki$_{1B}$ | Contriever-wiki$_{1B}$ | Revela-code$_{1B}$ | Contriever-code$_{1B}$ |
|---|---|---|---|---|
| **Apps** | 9.1 | 11.3 | 19.4 | **24.4** |
| **CosQA** | 22.6 | 24.3 | 30.2 | **32.4** |
| **ST2SQL** | **55.8** | 46.3 | 55.0 | 44.8 |
| **SN** | 54.2 | 49.8 | **64.0** | 41.5 |
| **SNCC** | 69.2 | 57.4 | **70.0** | 52.3 |
| **TransC** | 74.4 | 76.3 | **81.1** | 79.6 |
| **TransDL** | **34.7** | 33.8 | 34.2 | 30.8 |
| **SOQA** | 70.9 | 81.2 | 85.7 | **91.0** |
| **F-ST** | 70.6 | 65.5 | **76.2** | 66.0 |
| **F-MT** | **70.7** | 56.6 | 70.4 | 58.5 |
| **Mean** | 53.2 | 50.3 | **58.6** | 52.1 |

Table 10: Performance of unsupervised/self-supervised retriever models on BEIR datasets (nDCG@10, %). **Bold** marks the best score per dataset.

| Dataset | Revela-wiki$_{1B}$ | Contriever-wiki$_{1B}$ | Revela-code$_{1B}$ | Contriever-code$_{1B}$ |
|---|---|---|---|---|
| **ArguAna** | 44.6 | **48.9** | 46.9 | 42.0 |
| **ClimateFEVER** | 15.8 | **29.0** | 14.9 | 13.7 |
| **DBPedia** | 27.6 | **31.6** | 21.8 | 14.6 |
| **FEVER** | 61.7 | **68.6** | 51.4 | 41.2 |
| **FiQA2018** | 30.4 | 27.7 | **33.3** | 31.7 |
| **HotpotQA** | **56.0** | 47.1 | 46.4 | 17.6 |
| **NFCorpus** | 27.2 | **34.0** | 27.3 | 30.5 |
| **NQ** | **33.9** | 28.9 | 29.5 | 15.0 |
| **QuoraRetrieval** | 83.5 | 77.5 | **84.3** | 76.9 |
| **SCIDOCS** | 16.3 | **21.3** | 16.3 | 19.9 |
| **SciFact** | **71.9** | 63.9 | 66.4 | 61.5 |
| **TRECCOVID** | **60.1** | 55.5 | 54.2 | 39.5 |
| **Touche2020** | **25.7** | 17.4 | 21.8 | 15.2 |
| **Mean** | **42.7** | 42.4 | 39.6 | 32.3 |

domain tasks, Revela significantly surpasses Contiever, highlighting the advantage over traditional contrastive learning paradigm.

## C.2 ANALYSIS ON BATCH SIZES

Tab. 11 and Tab. 12 present the performance of Revela with different batch sizes on CoIR and BEIR, respectively. Generally, with a larger batch size, Revela's performance will be increased.

Table 11: Retrieval performance (nDCG@10, %) across the 10 CoIR tasks. We report three encoder sizes (135M, 500M, 1B) and three batch sizes (bs4, bs8, bs16).

| Dataset | Revela$_{0.1B}$ | | | Revela$_{0.5B}$ | | | Revela$_{1B}$ | | |
|---|---|---|---|---|---|---|---|---|---|
| | bs4 | bs8 | bs16 | bs4 | bs8 | bs16 | bs4 | bs8 | bs16 |
| **AppsRetrieval** | 4.8 | 6.4 | 8.2 | 11.8 | 16.5 | 20.5 | 17.1 | 16.2 | 19.4 |
| **CosQA** | 25.0 | 25.4 | 26.2 | 27.5 | 28.4 | 27.5 | 28.0 | 28.1 | 30.2 |
| **SyntheticText2SQL** | 42.8 | 44.0 | 45.7 | 50.9 | 52.5 | 53.7 | 53.4 | 51.4 | 55.0 |
| **COIRCodeSearchNetRetrieval** | 39.5 | 46.5 | 49.9 | 57.0 | 52.9 | 57.9 | 59.3 | 57.9 | 64.0 |
| **CodeSearchNetCCRetrieval** | 56.0 | 60.4 | 63.4 | 62.5 | 63.7 | 68.0 | 69.0 | 69.2 | 70.0 |
| **CodeTransOceanContest** | 63.4 | 68.1 | 70.9 | 71.6 | 74.0 | 77.6 | 75.6 | 77.7 | 81.1 |
| **CodeTransOceanDL** | 34.6 | 34.9 | 34.6 | 33.8 | 34.0 | 35.4 | 34.4 | 34.3 | 34.2 |
| **StackOverflowQA** | 57.1 | 62.8 | 69.2 | 73.8 | 76.6 | 82.5 | 77.6 | 82.5 | 85.7 |
| **CodeFeedbackST** | 59.0 | 61.2 | 63.8 | 74.2 | 74.9 | 74.5 | 74.8 | 75.5 | 76.2 |
| **CodeFeedbackMT** | 50.2 | 49.5 | 51.7 | 61.4 | 71.4 | 63.6 | 69.8 | 70.7 | 70.4 |
| **Mean** | 43.2 | 45.9 | 48.4 | 52.4 | 54.5 | 56.1 | 55.9 | 56.4 | 58.6 |

Table 12: Retrieval performance (nDCG@10, %) across 13 BEIR tasks for three encoder sizes and three batch sizes (`bs4`, `bs8`, `bs16`).

| Dataset | Revela$_{0.1B}$ | | | Revela$_{0.5B}$ | | | Revela$_{1B}$ | | |
|---|---|---|---|---|---|---|---|---|---|
| | bs4 | bs8 | bs16 | bs4 | bs8 | bs16 | bs4 | bs8 | bs16 |
| **ArguAna** | 36.7 | 38.5 | 39.0 | 35.4 | 41.0 | 41.1 | 43.0 | 47.9 | 44.6 |
| **ClimateFEVER** | 14.0 | 15.5 | 15.3 | 13.6 | 15.2 | 13.8 | 16.4 | 14.0 | 15.8 |
| **DBPedia** | 15.5 | 15.7 | 18.3 | 23.1 | 20.8 | 21.3 | 25.5 | 20.7 | 27.6 |
| **FEVER** | 27.0 | 29.5 | 33.7 | 47.3 | 46.1 | 51.1 | 57.7 | 51.2 | 61.7 |
| **FiQA2018** | 17.4 | 18.2 | 19.2 | 24.8 | 27.8 | 27.2 | 29.4 | 28.8 | 30.4 |
| **HotpotQA** | 32.7 | 33.3 | 38.5 | 45.7 | 46.0 | 50.6 | 51.0 | 54.6 | 56.0 |
| **NFCorpus** | 21.1 | 22.0 | 23.6 | 25.0 | 27.1 | 26.8 | 26.6 | 28.1 | 27.2 |
| **NQ** | 16.8 | 17.3 | 21.2 | 23.4 | 27.9 | 29.8 | 34.7 | 32.9 | 33.9 |
| **QuoraRetrieval** | 68.1 | 75.8 | 81.0 | 83.5 | 83.6 | 83.2 | 83.1 | 83.0 | 83.5 |
| **SCIDOCS** | 9.5 | 10.3 | 12.1 | 12.6 | 14.7 | 14.8 | 16.0 | 15.8 | 16.3 |
| **SciFact** | 52.4 | 56.3 | 57.9 | 61.1 | 62.4 | 66.0 | 65.5 | 67.9 | 71.9 |
| **TREC-COVID** | 58.8 | 58.1 | 60.6 | 62.3 | 64.1 | 58.4 | 67.1 | 68.8 | 60.1 |
| **Touche2020** | 14.6 | 12.7 | 12.4 | 21.1 | 22.4 | 19.9 | 26.7 | 24.7 | 25.7 |
| **Mean** | 29.6 | 31.0 | 33.3 | 36.8 | 38.4 | 38.8 | 41.7 | 41.4 | 42.7 |

## C.3  ANALYSIS ON THE SIZE OF LMS

Tab. 13 and Tab. 14 present an ablation study analyzing the impact of differently sized LMs.

Table 13: nDCG@10 (%) on BEIR for three model sizes (Revela$_{0.1B}$, Revela$_{0.5B}$, Revela$_{1B}$), with LM sizes in ascending order. Each triplet of columns highlights the highest value in **bold**.

| Dataset | Revela$_{0.1B}$ | | | Revela$_{0.5B}$ | | | Revela$_{1B}$ | | |
|---|---|---|---|---|---|---|---|---|---|
| | 0.1B | 0.5B | 1B | 0.1B | 0.5B | 1B | 0.1B | 0.5B | 1B |
| ArguAna | 36.9 | **39.9** | 39.0 | 38.5 | **41.8** | 41.1 | 41.1 | 43.0 | **44.6** |
| ClimateFEVER | 14.1 | **16.3** | 15.3 | 18.3 | **20.0** | 13.8 | 15.0 | **15.8** | 15.8 |
| DBPedia | 17.6 | 17.0 | **18.3** | **24.2** | 22.7 | 21.3 | 25.0 | 26.4 | **27.6** |
| FEVER | 25.5 | 31.3 | **33.7** | 53.0 | **55.7** | 51.1 | 58.7 | 56.7 | **61.7** |
| FiQA2018 | 16.9 | **19.3** | 19.2 | 26.6 | 24.9 | **27.2** | 29.3 | 28.4 | **30.4** |
| HotpotQA | **41.7** | 41.0 | 38.5 | **54.6** | 53.9 | 50.6 | 58.4 | **58.7** | 56.0 |
| NFCorpus | 19.7 | 22.2 | **23.6** | **27.4** | 27.4 | 26.8 | 26.4 | **28.7** | 27.2 |
| NQ | **22.3** | 21.0 | 21.2 | **34.5** | 32.2 | 29.8 | **36.8** | 34.9 | 33.9 |
| QuoraRetrieval | 79.1 | 79.7 | **81.0** | 83.4 | **83.5** | 83.2 | 82.4 | 83.3 | **83.5** |
| SCIDOCS | 10.8 | 11.3 | **12.1** | **15.4** | 15.1 | 14.8 | 15.9 | 16.0 | **16.3** |
| SciFact | 52.4 | 56.7 | **57.9** | 65.9 | 65.8 | **66.0** | 71.6 | **72.1** | 71.9 |
| TREC-COVID | 58.6 | **60.6** | 60.6 | **68.7** | 68.1 | 58.4 | **66.4** | 65.7 | 60.1 |
| Touche2020 | 11.2 | **13.1** | 12.4 | **31.1** | 24.2 | 19.9 | **28.7** | 26.9 | 25.7 |
| **Mean** | 31.3 | 33.0 | **33.3** | **41.6** | 41.2 | 38.8 | 42.7 | **42.8** | 42.7 |

Table 14: nDCG@10 (%) on CoIR for three model sizes (Revela$_{0.1B}$, Revela$_{0.5B}$, Revela$_{1B}$), with LM sizes in ascending order. **Bold** in each row indicates the best performance for each retriever size.

| Task | Revela$_{0.1B}$ | | | Revela$_{0.5B}$ | | | Revela$_{1B}$ | | |
|---|---|---|---|---|---|---|---|---|---|
| | 0.1B | 0.5B | 1B | 0.1B | 0.5B | 1B | 0.1B | 0.5B | 1B |
| AppsRetrieval | 5.9 | 5.2 | **8.2** | 18.2 | 15.0 | **20.5** | 17.9 | 15.7 | **19.4** |
| CosQA | 24.8 | 24.6 | **26.2** | 26.7 | **28.3** | 27.5 | 28.7 | **30.2** | 30.2 |
| SyntheticText2SQL | 43.9 | **46.2** | 45.7 | 50.8 | 52.0 | **53.7** | 52.1 | 53.0 | **55.0** |
| COIRCodeSearchNetRetrieval | 40.0 | 40.0 | **49.9** | 51.2 | 48.0 | **57.9** | 57.2 | 55.2 | **64.0** |
| CodeSearchNetCCRetrieval | 54.7 | 56.2 | **63.4** | 61.5 | 61.3 | **68.0** | 62.6 | 64.1 | **70.0** |
| CodeTransOceanContest | 67.0 | 67.6 | **70.9** | 74.0 | 74.0 | **77.6** | 76.8 | 76.8 | **81.1** |
| CodeTransOceanDL | 34.2 | **34.4** | 34.6 | 34.0 | **34.8** | 35.4 | 33.8 | **35.0** | 34.2 |
| StackOverflowQA | 67.1 | 65.4 | **69.2** | 79.5 | 77.0 | **82.5** | 82.2 | 80.9 | **85.7** |
| CodeFeedbackST | **64.8** | 64.9 | 63.8 | 73.3 | **73.9** | 74.5 | 75.2 | 75.0 | **76.2** |
| CodeFeedbackMT | **55.7** | 54.4 | 51.7 | **66.0** | 62.4 | 63.6 | 68.5 | **69.7** | 70.4 |
| **Mean** | 45.8 | 45.9 | **48.4** | 53.5 | 52.7 | **56.1** | 55.5 | 55.6 | **58.6** |

## C.4 MIXED-DOMAIN COMPOSITION

Tab. 15 and Tab. 16 report the performance of `Revela` trained on the mixture of the batches constructed from Wikipedia and code-related corpus, applied in § 4. We randomly sample 160,000 batches from both datasets to form the training data, and maintain all other experimental setups. Compared with Tab. 1 and Tab. 7, where `Revela` is trained separately in each domain, it can largely maintain the original performance when trained on the mixed-domain data. These results indicate `Revela`'s potential to generalize to diverse domains.

Table 15: Comparison of `Revela`'s performance (nDCG@10, %) across CoIR benchmark tasks for three LM sizes trained on the mixture of Wikipedia and code-related corpus. **Bold** indicates the best performance in each row.

| Dataset | $Revela_{0.1B}$ | $Revela_{0.5B}$ | $Revela_{1B}$ |
|---|---|---|---|
| **AppsRetrieval** | 5.9 | 17.2 | **17.5** |
| **CosQA** | 25.8 | **28.0** | 24.5 |
| **SyntheticText2SQL** | 46.0 | 51.5 | **55.1** |
| **COIRCodeSearchNetRetrieval** | 41.1 | 51.5 | **58.3** |
| **CodeSearchNetCCRetrieval** | 59.3 | 62.7 | **66.6** |
| **CodeTransOceanContest** | 66.3 | 76.4 | **78.6** |
| **CodeTransOceanDL** | 34.4 | **35.0** | 34.3 |
| **StackOverflowQA** | 66.0 | 79.4 | **82.2** |
| **CodeFeedbackST** | 64.5 | 74.6 | **75.6** |
| **CodeFeedbackMT** | 52.8 | 67.7 | **71.2** |
| **Mean** | 46.2 | 54.4 | **56.4** |

Table 16: Comparison of `Revela`'s performance (nDCG@10, %) across BEIR benchmark tasks for three LM sizes trained on the mixture of Wikipedia and code-related corpus. The highest value in each row is highlighted in **bold**.

| Dataset | $Revela_{0.1B}$ | $Revela_{0.5B}$ | $Revela_{1B}$ |
|---|---|---|---|
| **ArguAna** | 41.4 | 43.9 | **44.2** |
| **ClimateFEVER** | **16.2** | 16.1 | 15.3 |
| **DBPedia** | 18.8 | 21.7 | **25.4** |
| **FEVER** | 35.9 | 44.9 | **56.8** |
| **FiQA2018** | 21.2 | 28.3 | **34.6** |
| **HotpotQA** | 37.0 | 50.3 | **59.1** |
| **NFCorpus** | 24.5 | **28.7** | **28.7** |
| **NQ** | 19.7 | 27.4 | **34.5** |
| **QuoraRetrieval** | 82.2 | **85.7** | 85.2 |
| **SCIDOCS** | 11.7 | 15.8 | **16.1** |
| **SciFact** | 59.5 | 63.8 | **69.2** |
| **TRECCOVID** | 61.8 | 62.9 | **67.0** |
| **Touche2020** | 15.7 | 20.0 | **23.1** |
| **Mean** | 34.3 | 39.2 | **43.0** |

## C.5 OUT-OF-DOMAIN GENERALIZATION

To further validate `Revela`'s generalization, we trained it on Fineweb-edu (Penedo et al., 2024), a widely-used language modeling corpus.[8] Following the experimental setup from § 4.1, we trained for 320,000 batches and evaluated the models on the CoIR and BEIR benchmarks. The results in Tab. 17 and Tab. 18 show that `Revela` achieves competitive performance despite being trained on out-of-domain data. Notably, the $Revela_{0.5B}$ model scores 48.6% on CoIR, outperforming E5-PT (46.4%), which was trained on 270 million query-document pairs.

---

[8]https://huggingface.co/datasets/HuggingFaceFW/fineweb-edu

Table 17: Comparison of `Revela`'s performance (nDCG@10, %) across CoIR benchmark tasks for three LM sizes trained on an out-of-domain corpus, i.e., Fineweb-edu. **Bold** indicates the best performance in each row.

| Dataset | $\texttt{Revela}_{0.1B}$ | $\texttt{Revela}_{0.5B}$ | $\texttt{Revela}_{1B}$ |
|---|---|---|---|
| **AppsRetrieval** | 1.9 | 8.6 | **11.3** |
| **CosQA** | 25.8 | 26.0 | **26.6** |
| **SyntheticText2SQL** | 42.0 | **54.3** | 53.7 |
| **COIRCodeSearchNetRetrieval** | 27.8 | 38.9 | **46.2** |
| **CodeSearchNetCCRetrieval** | 47.0 | 60.5 | **61.7** |
| **CodeTransOceanContest** | 60.2 | 68.6 | **75.2** |
| **CodeTransOceanDL** | 33.4 | 33.5 | **33.8** |
| **StackOverflowQA** | 50.6 | 63.1 | **69.0** |
| **CodeFeedbackST** | 52.7 | 70.4 | **70.9** |
| **CodeFeedbackMT** | 38.7 | 61.8 | **66.9** |
| **Mean** | 38.0 | 48.6 | **51.5** |

Table 18: Comparison of `Revela`'s performance (nDCG@10, %) across 13 BEIR tasks for three LM sizes trained on an out-of-domain corpus, i.e., Fineweb-edu. **Bold** indicates the best performance in each row.

| Dataset | $\texttt{Revela}_{0.1B}$ | $\texttt{Revela}_{0.5B}$ | $\texttt{Revela}_{1B}$ |
|---|---|---|---|
| **ArguAna** | 36.4 | 36.9 | **38.7** |
| **ClimateFEVER** | **13.9** | 12.7 | 12.3 |
| **DBPedia** | 17.2 | 20.2 | **24.5** |
| **FEVER** | 30.5 | 45.0 | **53.9** |
| **FiQA2018** | 20.1 | 24.9 | **30.3** |
| **HotpotQA** | 29.7 | 47.5 | **54.2** |
| **NFCorpus** | 26.8 | **30.1** | 29.7 |
| **NQ** | 17.4 | 24.8 | **31.0** |
| **QuoraRetrieval** | 81.8 | **83.6** | 82.4 |
| **SCIDOCS** | 12.6 | 15.6 | **16.8** |
| **SciFact** | 59.3 | 65.1 | **68.6** |
| **TRECCOVID** | 65.5 | **66.3** | 63.3 |
| **Touche2020** | 15.8 | 20.9 | **23.0** |
| **Mean** | 32.9 | 38.0 | **40.7** |

## C.6 LM PERFORMANCE AFTER REVELA

We evaluate the original LLaMA-3.2-1B and the model co-trained with the retriever (LLaMA-3.2-1B backboned on Wikipedia in § 4.1) on seven tasks, including ARC (Clark et al., 2018), COPA (Roemmele et al., 2011), OpenBookQA (Mihaylov et al., 2018), PIQA (Bisk et al., 2020), RTE (Dagan et al., 2005), WiC (Pilehvar & Camacho-Collados, 2019), and Winogrande (Sakaguchi et al., 2020). As shown in Tab. 19, LM's performance has been greatly preserved.

Table 19: Performance (Accuracy, %) of the LM before and after `Revela` on various benchmarks.

| Model | ARC | COPA | OpenBookQA | PIQA | RTE | WiC | WinoGrande | Avg. |
|---|---|---|---|---|---|---|---|---|
| LLaMA-3.2-1B | 28.0 | 77.0 | 25.0 | 73.0 | 59.5 | 47.5 | 57.5 | 52.5 |
| LM-`Revela` | 29.0 | 74.0 | 25.5 | 71.0 | 58.5 | 52.0 | 55.5 | 52.2 |

## C.7 COMPARISON BETWEEN REVELA AND REPLUG

While both `Revela` and REPLUG involve retrievers and LMs, their core motivations differ. REPLUG distills relevance signals from the perplexity of frozen, off-the-shelf LMs. In contrast, `Revela` proposes a broader paradigm: to learn retrievers as an integral part of language modeling, rather than as a post hoc add-on. From a technical perspective, `Revela` also outperforms REPLUG regarding efficiency.

For a batch of $B$ documents (each with sequence length $L$), the cost of modeling inter-document relationships differs significantly between REPLUG and `Revela`:

- REPLUG: To compute pairwise relevance between all document pairs using LM perplexity requires processing each pair independently. Each pair is concatenated (length $2L$) and passed through the LM, resulting in a forward cost of $\mathcal{O}(B^2 \times (2L)^2)$ for all $B(B-1)/2$ pairs.
- Revela: As detailed in § 3.4, uses in-batch attention with concatenated inputs. By processing all $B$ documents jointly in a single forward pass with cross-attention, it computes inter-document relevance with a cost of $\mathcal{O}(B \times (2L)^2)$. Including backward passes (which scale similarly), the total cost remains linear in batch size.

Therefore, when using a reasonably large batch size (e.g., 16 as in our setup), Revela learns inter-document relevance more efficiently than REPLUG, due to its **linear** scaling in batch size compared to REPLUG's **quadratic** cost.

## D    USE OF LLMs

In preparing this manuscript, we employed LLM for general assistance. Its use focuses on improving the clarity and grammar of the text and helping generate the code used to create figures (e.g., matplotlib) or LaTeX tables.

