# OpenReview forum: "Revela: Dense Retriever Learning via Language Modeling"
_ICLR.cc/2026/Conference — ICLR 2026 Oral_

### Official Review · Reviewer_CDCx · 2025-10-23

**Soundness:** 3
**Presentation:** 2
**Contribution:** 3
**Rating:** 6
**Confidence:** 3

**Summary:**

This paper introduces Revela, a self-supervised retrieval model training method that employs a language modeling task. Revela incorporates in-batch attention, allowing the model to consider both the current sequence and the contexts of other sequences within the same batch when predicting the next token, thereby modeling semantic dependencies between documents. This approach eliminates the need for manually labeled query-document pairs and directly trains the retriever using the modeling task, significantly improving the training efficiency and effectiveness of the model.

**Strengths:**

1. This paper demonstrates excellent performance across a variety of retrieval tasks with different types, showcasing its versatility.
2. The method requires a smaller batch size. During training, the batch size used in this paper was 8, which is much smaller than other methods that employ contrastive learning for training. This significantly reduces the forest traversal cost. Additionally, the experiments clearly reveal the scaling law between retrieval performance and batch size.
3. There is no need for labeled q-d pairs as training data, and the model can be directly trained in an unsupervised manner on large-scale text datasets.

**Weaknesses:**

1. This paper compares retrieval models that include both encoder-only and decoder-only models, and it also conducts comparisons based on different model sizes. However, there are still some fairness issues in the comparison.  The original RePlug paper uses query-document pairs for KL training in the RAG setting, while this paper only uses document alone in the reproduction and comparison. This might not be a fair comparison. Could training on the q-d pair data mentioned in RePlug, such as the Pile training data or other q-d datasets like MS MARCO, provide a fairer comparison? RePlug is this paper's main baseline.

**Questions:**

1. In this paper, it is not explicitly mentioned whether the representation used during inference is from the last token or if mean pooling over all token representations is applied.

2. Is the backbone model used in REPLUG also Llama 3.2?

3. In addition to NDCG@10, could you provide results for other metrics such as MRR and recall?

4. It seems that no ablation experiments are designed in this paper.

---

> ### Author Response · Authors · 2025-11-21
> **Thank you for the review! - Part 1**
>
> We sincerely thank you for recognizing Revela’s excellent performance and its fresh perspective of retriever learning without query-document pairs. We would address your concerns as follows.
>
> > W1: **Comparison with RePlug**
>
> Revela is designed to learn retrievers from an unlabeled corpus. While RePlug was originally demonstrated using query-document pairs in a RAG setting, its core mechanism is **to distill the frozen LM's perplexity** into a similarity score between two pieces of text. This objective is *not* inherently restricted to query–document pairs. In our main comparison, Revela and RePlug therefore use the same dataset, the same retriever architecture, and the same reference language model. Under these matched conditions, we believe the comparison is fair.
>
> That said, we agree it is valuable to further evaluate Revela in a query-document setting, as suggested. We therefore conduct additional experiments on MS MARCO (`Tevatron/msmarco-passage-aug`), training both Revela and RePlug with `Llama3.2-1B` as both the retriever backbone and the reference LM. For batch construction, we follow the standard supervised setup: each batch consists of one query, one positive passage, and 14 negative passages. All other training configurations are kept identical to those used in the main paper.
>
> We find the `Revela-msmarco-1b` still outperforms `Replug-msmarco-1b` by a significant margin, and also surpasses `Revela-wiki-1b` reported in Table 6 in the paper. We suspect that Replug is **limited by the imperfect calibration** of LM [1] when perplexity is used as a similarity signal. In contrast, Revela remains effective and even stronger when trained on labeled query-document pairs, indicating that its advantages extend beyond the purely unsupervised setting and supporting the fairness and robustness of our main comparison.
>
> | Model              | ArguAna | ClimateFEVER | DBPedia | FEVER | FiQA2018 | HotpotQA | NFCorpus | NQ   | QuoraRetrieval | SCIDOCS | SciFact | TRECCOVID | Touche2020 | Mean |
> |--------------------|---------|-------------|---------|-------|----------|----------|----------|------|----------------|---------|---------|-----------|------------|------|
> | `Revela-msmarco-1b`  | 52.5    | 17.4        | 34.6    | 63.0  | 35.0     | 61.8     | 35.2     | 41.7 | 86.7           | 19.5    | 75.0    | 73.0      | 21.1       | 47.4 |
> | `Revela-wiki-1b`     | 44.6    | 15.8        | 27.6    | 61.7  | 30.4     | 56.0     | 27.2     | 33.9 | 83.5           | 16.3    | 71.9    | 60.1      | 25.7       | 42.7 |
> | `Replug-msmarco-1b`  | 39.5    | 17.0        | 26.3    | 55.9  | 28.1     | 51.1     | 25.7     | 26.1 | 89.1           | 21.2    | 71.1    | 37.0      | 17.8       | 38.9 |
>
> [1] A Survey of Confidence Estimation and Calibration in Large Language Models, NAACL 2024

---

> ### Author Response · Authors · 2025-11-21
> **Thank you for the review! - Part 2**
>
> > Q1: **Representation for retrieval**
>
> As mentioned in L271, `<eos>` is appended to the end of the sequences and used for representation.
>
> > Q2: **REPLUG backbone**
>
> Yes, REPLUG uses the same backbone with Revela. For example, REPLUG uses `Llama3.2-1B` and `Llama3.2-3B` in Table 1.
>
> > Q3: **Other metrics beyond nDCG@10**
>
> Following your suggestion, we include both MRR@10 and Recall@10 corresponding to the results in Table 1. Evaluated on other metrics, Revela also exhibits excellent performance over RePlug and supervised models such as E5-Mistral-7B-Instruct.
>
> **MRR@10**
>
> | Dataset | BM25 | E5-PT | Revela-0.5B | BGE† | REPLUG-1B | Revela-1B | REPLUG-3B | Revela-3B | E5-Mistral |
> |--------|------|--------|-------------|------|-----------|-----------|-----------|-----------|-------------|
> | Apps   | 4.1  | 9.1    | 18.0        | 12.8 | 12.3      | 16.9      | 15.4      | **23.5**  | 20.8        |
> | CosQA  | 9.1  | 20.2   | 21.7        | 19.8 | 14.9      | 21.8      | 14.3      | 21.4      | **24.6**    |
> | ST2SQL | 18.2 | 48.9   | 50.4        | 51.7 | 61.5      | 52.7      | 53.1      | 52.0      | **65.4**    |
> | SN     | 37.9 | 32.3   | 54.2        | 54.9 | 47.4      | 60.2      | 50.3      | 59.1      | **63.8**    |
> | SNCC   | 49.9 | 46.4   | 63.8        | 49.5 | 58.2      | **65.7**  | 60.4      | 64.7      | 60.4        |
> | TransC | 43.9 | 52.6   | 75.5        | 59.1 | 76.2      | 78.4      | 78.0      | **80.7**  | 77.4        |
> | TransDL| 17.0 | 17.5   | **20.3**    | 15.6 | 18.0      | 18.5      | 19.3      | 20.1      | 14.5        |
> | SOQA   | 67.8 | 84.9   | 80.2        | 77.9 | 75.3      | 83.8      | 75.9      | 86.5      | **89.4**    |
> | F-ST   | 62.3 | 65.5   | 70.2        | 64.3 | 65.5      | 72.1      | 65.1      | **74.8**  | 71.5        |
> | F-MT   | 56.5 | 42.9   | 60.6        | 43.8 | 55.6      | 67.6      | 58.9      | **70.1**  | 32.3        |
> | Mean   | 36.7 | 42.0   | 51.5        | 44.9 | 48.5      | 53.8      | 49.1      | **55.3**      | 52.0        |
>
>
> **Recall@10**
>
> | Dataset | BM25 | E5-PT | Revela-0.5B | BGE† | REPLUG-1B | Revela-1B | REPLUG-3B | Revela-3B | E5-Mistral |
> |--------|------|--------|-------------|------|-----------|-----------|-----------|-----------|-------------|
> | Apps   | 6.9  | 15.6   | 28.6        | 21.1 | 20.6      | 27.3      | 24.9      | **36.4**      | 32.1        |
> | CosQA  | 26.0 | 47.6   | 48.4        | 46.6 | 34.8      | 51.6      | 37.0      | 50.0      | **53.2**        |
> | ST2SQL | 45.4 | 80.4   | 84.2        | 78.5 | 86.6      | 87.3      | 87.4      | 86.7      | **95.8**        |
> | SN     | 50.2 | 44.5   | 69.4        | 69.1 | 61.5      | 76.0      | 64.5      | 73.9      | **78.7**        |
> | SNCC   | 66.4 | 63.5   | 81.0        | 66.9 | 76.9      | **83.6**      | 79.9      | 82.8      | 78.6        |
> | TransC | 60.2 | 67.9   | 84.6        | 73.8 | 86.4      | 89.6      | 87.3      | **91.0**      | 90.5        |
> | TransDL| 80.6 | 73.3   | 82.2        | 70.6 | 81.7      | 81.1      | **82.8**      | 81.1      | 78.3        |
> | SOQA   | 77.8 | 93.4   | 89.5        | 88.8 | 84.9      | 91.6      | 85.0      | 93.8      | **95.8**        |
> | F-ST   | 80.1 | 85.1   | 87.6        | 84.6 | 83.4      | 89.1      | 83.0      | **91.2**      | 91.0        |
> | F-MT   | 67.8 | 56.6   | 72.9        | 60.6 | 67.9      | 79.2      | 70.9      | **81.9**      | 49.6        |
> | Mean | 56.1 | 62.8 | 72.9 | 66.0 | 68.5 | 75.6 | 70.3 | **76.9** | 74.4 |
>
> Q4: **Ablation study**
>
> Revela proposes a fresh perspective of retriever learning. To validate Revela’s effectiveness and generalization, we include ablation studies from both model and data dimensions in the manuscript.
>
> - From the **model** aspect, we include different families and scales of LMs for both retriever backbones and retriever in Sec. 5.
>
> - For the **data** aspect, we study the effect of batch sizes (L407) and other configurations of data composition, including mixed-domain training corpora (L450 and Appendix C.3) and out-of-domain corpora (L453 and Appendix C.5).
>
> Thank you again for your valuable suggestions and questions, which have helped us further improve the quality of our work. We hope that we have satisfactorily addressed all of your concerns.
>
> Best regards,
>
> Authors of Revela

---

> > ### Comment · Reviewer_CDCx · 2025-11-27
> > **Response to Author**
> >
> > Thank you for the response, which addressed my concerns. I will adjust my rating.

---

> > > ### Author Response · Authors · 2025-11-28
> > >
> > > Dear Reviewer CDCx,
> > >
> > > We are glad that our response has addressed your concerns, and we truly appreciate your recognition of Revela.
> > >
> > > Best regards,
> > >
> > > Authors of Revela

---

### Official Review · Reviewer_PRP5 · 2025-10-23

**Soundness:** 4
**Presentation:** 3
**Contribution:** 3
**Rating:** 6
**Confidence:** 4

**Summary:**

**Motivation:** Dense retrievers are important, but training them usually needs labeled query–document pairs, which are costly and scarce in specialized domains. The paper asks if we can train a retriever in a self-supervised way by treating retrieval like language modeling, where dependencies among chunks of text play the role that token dependencies play in next token prediction.

**Approach:** Revela trains a retriever together with a decoder-only LM by adding an in-batch attention path that lets each sequence attend to other sequences, with those cross-sequence attention weights set by the retriever’s similarity scores, and both parts optimized through next token prediction. Documents are split into chunks and batched so the model can learn cross-document links; similarities are temperature-normalized cosine scores that modulate the in-batch attention inside each transformer block.

**Key Results:** Across CoIR, BRIGHT, and BEIR, Revela beats or matches strong baselines while using only raw text: on CoIR it surpasses E5-Mistral-7B-Instruct by 2.8 points and outperforms E5-PT by 9.7 points at smaller scales. On BRIGHT it is on par with large supervised and API models, and on BEIR it reaches the unsupervised state of the art with about 1000× less data and 10× less compute, with gains that grow with model and batch size.

**Strengths:**

**Unified objective without labels:** The method removes the need for query–document pairs by turning retrieval into language modeling. I like the idea of using NTP for training retriever as it can provide fine-grained supervision compared to InfoNCE. There is a related work of “REPLUG” which also uses supervision from language modeling to train retriever. But the approach from this paper: adding a simple in-batch attention path to a standard decoder-only stack and learning both the retriever and LM with next-token loss has its novelty.

**Strong results across domains:** It beats a larger supervised E5-Mistral-7B model on CoIR, stays competitive with top supervised and API systems on BRIGHT, and matches E5-PT on BEIR while using about 1000× less data and 10× less compute. The wins hold across different topics and corpus types, which suggests good robustness and transfer.

**Good scaling and clear ablations:** Performance improves as you increase batch size, retriever size, and often LM size. Ablations show the cross-sequence attention and similarity weighting matter, and the model outperforms a matched Contriever baseline with the same backbone. Training on mixed domains keeps quality and can even help.

**Weaknesses:**

**Applicability**: Adoption may be harder than plug-and-play methods like REPLUG. Revela adds an extra in-batch attention path inside decoder blocks and uses retriever similarities as attention weights, which means touching the LM internals and maintaining custom masking. Even if the code changes are “minimal,” such modifications increase maintenance and may impact speed or memory in non-obvious ways.

**Batch-composition dependence:** The method conditions next-token prediction on *other documents in the same batch*. This can make results sensitive to how batches are formed or interleaved, and the paper does not deeply analyze robustness to batch construction.

**Resources**: Compared to approaches using single retriever with InfoNCE loss, training will probably likely to require high resources compared to existing works, as it requires optimizing both the retrievers and LLMs. This can make the approach use less batch size given same compute budget.

**Questions:**

1. Can we keep the LM frozen and train only the retriever while keeping most of the gains?
2. How did you deal with the causal mask when using LLM as the retriever backbone?
3. How robust is performance to batch composition choices, such as mixing documents from different topics or changing the interleaving pattern across chunks?

---

> ### Author Response · Authors · 2025-11-21
>
> We sincerely appreciate your recognition of Revela’s ability to leverage unlabeled data, its strong performance across domains, its scaling capacity, and the thoroughness of our ablation study. We would address your concerns in detail below.
>
> > W1: **Applicability**
>
> Compared with plug-and-play methods that keep the LM completely frozen, Revela consistently outperforms RePlug across model scales (0.1B–3B) and benchmarks (BEIR, CoIR, BRIGHT). Moreover, as discussed in L458 and Appendix C.7, Revela is **more efficient** for retriever learning than RePlug. We therefore view Revela’s applicability from both effectiveness and efficiency: its modest internal modifications enable substantially better retrievers at lower training cost.
>
> Regarding implementation complexity, Revela does require a **small** modification to the decoder attention, i.e., adding an in-batch path that uses retriever similarities with a custom mask. However, these changes are localized to the attention module and do not alter the overall model architecture, optimizer, or training loop. To lower the adoption barrier, we provide the adapted code within the standard `transformers`  (a modified `modeling_llama.py`) in our submission and will release it publicly as a reference implementation. With this integration, we expect the community to maintain the modifications alongside existing model variants and to reuse Revela as a small, self-contained plugin for effective retriever learning.
>
> > W2 & Q3: **Batch construction**
>
> Revela adopts a relatively relaxed strategy for batch construction. As shown in L261 and Fig. 7 in Appendix B.2, a single batch can include chunks originating from different documents and topics. This design makes Revela easy to apply across diverse domains and scenarios without requiring carefully curated batches.
>
> > W3: **More computational resources**
>
> We acknowledge that co-training both the retriever and the LM makes Revela more resource-intensive than methods that train a retriever with an InfoNCE loss. However, we emphasize that Revela delivers substantial benefits for a **modest** additional cost:
>
> - **Data construction**: Revela completely **removes** the need for query–document pairs, addressing a long-standing bottleneck in retriever training.
>
> - **Performance**: On unsupervised data, Revela consistently outperforms InfoNCE-style training in both in-domain performance and out-of-domain generalization (see Section starting at L389).
>
> - **Applicability with smaller LMs**: As shown in Fig. 5, Revela remains competitive even when co-trained with 0.1B-parameter LMs, incurring only a small overall computational overhead.
>
> For the retrieval community, Revela thus offers a practical way to better leverage LMs and large unlabeled corpora, trading a modest increase in compute for stronger performance and broader applicability.
>
> >**Q1: Can LM be frozen?**
>
> From a technical perspective, it is necessary to make the LMs **trainable**, as they are not originally designed to model a sequence while attending to other sequences within the same batch. Freezing the LMs can lead to significant issues: most notably, the misalignment of **position embeddings**, which disrupts the attention calculation across sequences since all the sequences start from position 0.
>
> Empirically, we run `Revela-wiki-1b` with LM frozen while keeping all other setups and using 1b LM as the reference model. As shown below, its performance on BEIR significantly dropped compared with `Revela-wiki-1b`.
>
> | Model                               | ArguAna | C-FEVER | DBPedia | FEVER | FiQA | HotpotQA | NFCorpus |   NQ   | Quora | SCIDOCS | SciFact | TREC-C | Touche | Mean  |
> |------------------------------------|--------:|--------:|--------:|------:|---------:|---------:|---------:|-------:|------:|--------:|--------:|-------:|--------:|------:|
> | `Revela-wiki-1b` (Frozen LM)                 |  13.5   |   5.8   |  11.7   | 26.0  |   8.4    |   7.5    |  10.1    |  10.7  | 52.6  |   5.2   |  17.6   | 23.5   |  17.8   | 16.2  |
> | `Revela-wiki-1b`               |**44.6**| **15.8**|  **27.6**   | **61.7**  | **30.4** |  **56.0**    |  **27.2**    |  **33.9**  | **83.5**  |  **16.3**   | **71.9**    | **60.1**   | **25.7**   | **42.7** |
>
> >**Q2: Casual mask in retriever**
>
> We maintain the casual mask in LM-base retrievers and use `<eos>` as the representation, as mentioned in L271.
>
> Thank you once again for your insightful suggestions and questions, which have contributed to improving the quality of our work. We hope that our responses have fully addressed your concerns.
>
> Regards,
>
> Authors of Revela

---

> > ### Comment · Reviewer_PRP5 · 2025-11-24
> > **Response to Authors**
> >
> > Thank you for the response. It addresses most of my concerns, and I have adjusted the scores accordingly.

---

> > > ### Author Response · Authors · 2025-11-24
> > >
> > > Dear Reviewer PRP5,
> > >
> > > Thank you for your encouraging feedback. We are glad that our response has addressed your concerns, and we truly appreciate your willingness to raise the score.
> > >
> > > Regards,
> > >
> > > Authors of Revela

---

### Official Review · Reviewer_U5J6 · 2025-10-31

**Soundness:** 3
**Presentation:** 3
**Contribution:** 3
**Rating:** 6
**Confidence:** 3

**Summary:**

Revela proposes a novel self-supervised framework that jointly trains a dense retriever and a language model by embedding retriever similarity scores as in-batch attention weights within transformer blocks.
This allows the model to learn retrieval ability directly from next-token prediction without any labeled query-document pairs, achieving strong performance on multiple retrieval benchmarks with far less data and computation.

**Strengths:**

- Revela proposes a novel self-supervised framework that jointly trains a dense retriever and a language model by embedding retriever similarity scores as in-batch attention weights within transformer blocks.
This allows the model to learn retrieval ability directly from next-token prediction without any labeled query-document pairs, achieving strong performance on multiple retrieval benchmarks with far less data and computation.
- The use of retriever similarity as in-batch attention weights seamlessly embeds retrieval into standard Transformer computation with minimal architectural changes.

**Weaknesses:**

- The motivation for using next-token prediction is unclear. The authors need to provide a detailed explanation of why this training objective can enhance the retriever’s capability.
- The In-batch Attention section is somewhat confusing. It states that In-batch Attention consists of two parts — Standard Self-Attention and In-batch Attention — but within In-batch Attention itself, there is another self-attention output s. I suggest the authors restructure this description for greater clarity.
- Could the authors provide a detailed explanation of the computation steps for In-batch Attention? My understanding is that it involves three internal computations (outputs e, s, and b), yet Figure 2 shows only a single attention map (indicating the computation is fused?)

**Questions:**

The questions have been presented above.

---

> ### Author Response · Authors · 2025-11-21
> **Thanks for your review!**
>
> We would sincerely thank you for your recognition of our work. We would explain the motivation of NTP and thereby Revela’s mechanism as follows.
>
> > W1: **Motivation of using NTP (language modeling) for retriever learning**
>
> NTP is a self-supervised objective that trains language models to capture dependencies among tokens within a sequence. In Revela, we extend this idea from *token-level* dependencies to *document-level* (or chunk-level) dependencies, and this is precisely what allows NTP to improve the retriever.
>
> Concretely: A dense retriever is expected to model relationships among *chunks of tokens* (i.e., documents or passages), capturing higher-level dependencies such as semantic relatedness, causal links, or shared factual content.
>
> In Revela, we couple the retriever and the LM by letting **retriever-computed similarities** modulate an **in-batch attention** mechanism during NTP (see Eq. (2), Sec. 3.1–3.3). For a sequence $D_i$, the NTP objective is conditioned not only on its own prefix $x_i^{<l}$ but also on all other documents $\{D_j\}_{j \neq i}$ in the batch, weighted by the retriever’s similarity scores $\text{Sim}(D_i, D_j)$.
>
> Intuitively, as shown in Fig 1,  to **lower the perplexity** of the LM on sequence $D_i$, the model learns to:
>
> 1. Attend more strongly to other sequences $D_j$ that contain information useful for predicting the next token in $D_i$; and
> 2. Down-weight unhelpful or irrelevant sequences.
>
> Therefore, in Revela, the retriever can be learned in a scalable, unsupervised way where unlabelled corpora can be used.
>
> > W3: **Explanation of computation for in-batch attention**
>
> `TL;DR`: In practice, this is implemented as a **single fused attention operation** with a specially designed block attention mask over the duplicated tokens. Conceptually, however, the outputs $e$, $s$, and $b$ correspond to different quadrants of that single attention map in Fig. 2 (upper-left for $e$, bottom-right for $s$, and bottom-left for $b$). This is why Fig. 2 shows only one attention map even though there are three internal computations.
>
> In detail, we would explain the computation of in-batch attention in a reverse order, starting from Eq. 7.
>
> Given the objective described above, i.e., NTP attending to both prior tokens in the same sequence and to other sequences, we need to combine self-attention $s$ and cross-document attention $b$ to form the in-batch attention output $h$ in Eq. 7. In Fig. 2:
> - The **bottom-right** quarter corresponds to the self-attention part $s$, with a **causal mask** inside the same sequence.
> - The **bottom-left** quarter corresponds to the cross-document part $b$, with **full attention** from the current sequence to other documents in the batch.
>
> Please note that although $s$ is a self-attention term, its input is $h^{l-1}$ from the previous layer, which **already** contains in-batch information. Therefore, we also need to independently compute a **pure** self-attention representation for a single sequence, denoted as $e$. The attention of \(e\) corresponds to the **upper-left** quarter in Fig. 2.
> As mentioned in Sec. 3.2 and 3.4, we implement this by **duplicating the sequence tokens** when inputting to the LM, i.e., $[t_1, t_2, \ldots, t_n, \; t_1, t_2, \ldots, t_n]$.
> - The **first copy** is used for standard self-attention computation, i.e., to obtain $e$.
> - The **second copy** is used for in-batch attention, i.e., to compute $s$ (self-attention within the second copy) and $b$ (cross-document attention to other sequences, weighted by similarity).
>
> Thank you again for your valuable suggestions and questions, which have helped us further improve the quality of our work. Following your suggestion, we will further clarify our description of the algorithm (Sec. 3) in the final manuscript (**W2**). We hope that we have satisfactorily addressed all of your concerns.
>
> Regards,
>
> Authors of Revela

---

> > ### Comment · Reviewer_U5J6 · 2025-11-25
> >
> > Thank you for clarification, which addressed my concerns. I am adjusting my rating to be more positive.

---

> > > ### Author Response · Authors · 2025-11-26
> > >
> > > Dear Reviewer U5J6,
> > >
> > > Thank you for your effort and time! We are glad that our explanation on Revela addressed your concerns, and we truly appreciate your willingness to raise the score.
> > >
> > > Regards,
> > >
> > > Authors of Revela

---

### Official Review · Reviewer_ojFX · 2025-11-04

**Soundness:** 4
**Presentation:** 4
**Contribution:** 4
**Rating:** 8
**Confidence:** 5

**Summary:**

This paper introduces a self-supervised framework that jointly trains a dense retrieval model along with a LLM using NTP loss and in-batch, cross document attention. The authors posit that NTP loss, when conditioned on in-batch chunks from different documents, weighted by the similarity scores of the retriever, provides strong supervision for measuring similarity among documents.

Experiments demonstrate that a dense retriever trained with <400k batches (~1000x less than modern retriever training datasets) out-performs strong baselines and enterprise API based models for standard tasks (BEIR) and coding (CoIR) among other sets.

**Strengths:**

1. The approach of using NTP paradigm to "distill" similarity signals into a dense retriever along with he in-batch attention and weighting by similarity scores method, are interesting and novel ideas.
2. Improved scalability and calibration compared to quadratic baseline (pairwise distillation).
3. Great experimental design including selection of training data and baselines.
4. Thorough ablation study showcases generalization, effect of batch size, LLM base performance, mixing training corpora among others. These studies provide a strong baseline for future exploration.
5. Overall, a very well written paper and easy to understand.

**Weaknesses:**

1. Training data creation methodology and statistics are underspecified. It would be helpful to understand how the filtering was done and what the "handcrafted rules" L249 are. Similarly, while constructing a batch (following example in Appendix B.2), how were the topics chosen?

**Questions:**

1. Given the joint training, what are the training dynamics and stability of the system? Does the retriever or the LM converge significantly faster?

---

> ### Author Response · Authors · 2025-11-21
> **Thank you for the review!**
>
> We sincerely appreciate your recognition of our work, particularly regarding its novelty, performance, experimental design, and presentation. We address your concerns as follows:
> - **Training data**: We would kindly point out that L249 describes the difficulty of data construction in E5 [1], which requires massive corpus (1.3B noisy pairs) and complicated filtering operations. Instead, Revela is convenient on data construction: taking Wikipedia as example, we split the passages from `Tevatron/wikipedia-nq` to sentences with NLTK, and make sure each chunk is now longer than 120 words. One batch can include chunks from multiple Wikipedia passages as mentioned in L261. For Wikipedia, we construct 320,000 batches from 339,409 Wikipedia passages. The topics (passages) are **randomly** sampled from the `Tevatron/wikipedia-nq`.
>
> - **Training dynamics**: We exponentially evaluate `Revela-1b` checkpoints over 11k training steps on CoIR. As shown in the table below, training is stable and performance improves steadily: scores become competitive by step 1600, and later checkpoints (e.g., 3200 and Revela) deliver the best mean results and dominate most individual tasks, with only marginal gains beyond 3200, indicating that the model has largely converged.
>
> | Task    | 100 | 200 | 400 | 800 | 1600 | 3200 | 6400 | Revela-1b-code |
> |---------|-----|-----|-----|-----|------|------|------|--------|
> | Apps    | 9.0 | 12.8 | 9.1 | 12.2 | 18.6 | 19.0 | 17.7 | **19.4** |
> | CosQA   | 21.3 | 21.5 | 23.9 | 22.9 | 27.4 | 27.1 | 26.9 | **29.6** |
> | ST2SQL  | 42.5 | 50.1 | 48.0 | 56.0 | 55.3 | 55.2 | 52.7 | **56.4** |
> | SN      | 48.1 | 51.4 | 46.4 | 50.9 | 58.6 | 59.7 | 61.6 | **64.0** |
> | SNCC    | 64.1 | 60.9 | 63.3 | 61.7 | 64.0 | 67.4 | 67.1 | **70.0** |
> | TransC  | 69.6 | 70.4 | 64.9 | 75.5 | 79.1 | 75.9 | 75.4 | **81.1** |
> | TransDL | 34.7 | 35.4 | 34.8 | **35.7** | 34.9 | 35.2 | 34.7 | 34.6 |
> | SOQA    | 71.6 | 77.3 | 68.2 | 69.2 | 75.9 | 81.6 | 81.5 | **85.7** |
> | F-ST    | 68.9 | 72.5 | 68.0 | 73.9 | 75.3 | **76.6** | 75.6 | 76.3 |
> | F-MT    | 67.4 | 61.1 | 63.1 | 70.0 | 66.4 | **74.5** | 67.4 | 70.4 |
> | Mean    | 49.7 | 51.3 | 49.0 | 52.8 | 55.5 | 57.2 | 56.1 | **58.7** |
>
> Thank you again for your valuable question on training dynamics, which has helped us better explore and understand Revela. We hope that we have satisfactorily addressed both of your concerns.
>
> Regards,
>
> Authors of Revela
>
> [1] Text Embeddings by Weakly-Supervised Contrastive Pre-training

---

### Author Response · Authors · 2025-12-03
**General Response**

Dear AC and Reviewers,

We are sincerely grateful for the time and effort you have devoted to reviewing our paper, especially during the recent incident. We also deeply appreciate the **positive** recognition from **all** reviewers and we are glad that our responses have successfully addressed the concerns and received explicit confirmation from Reviewers U5J6, PRP5, and CDCx, who all raised the scores.

Below, we provide a concise summary of Revela’s main contributions and strengths, as well as our responses to the reviewers’ comments.

### **Contributions**

Revela introduces a **self-supervised** framework for retriever learning that goes beyond contrastive learning by jointly training an LM and a retriever. It directly connects **next-token prediction (NTP)**, the standard language modeling objective, to retriever learning via in-batch attention, enabling retrieval to be learned from **raw text without requiring query-document pairs**. Revela outperforms and matches some **supervised** models and **proprietary APIs** on advanced retrieval tasks, such as code retrieval and reasoning-intensive retrieval.

### **Strengths Recognized by the Reviewers**

- **Novelty (ojFX, U5J6, PRP5)**: A novel use of in-batch attention to seamlessly associate retriever learning with the objective of language modeling, i.e., NTP.
- **Retriever learning from raw text (U5J6, PRP5, CDCx)**: The retriever is trained directly on raw text, instead of relying on pre-constructed query-document pairs.
- **Strong performance (ojFX, U5J6, PRP5, CDCx)**: Revela achieves strong performance across general, code, and reasoning-intensive retrieval tasks (Sec 4.2).
- **Robustness and domain generalization (ojFX, U5J6, PRP5, CDCx)**: When trained on data from one domain, Revela still performs robustly when applied to other domains (L389 and L450).
- **Scalability (ojFX, PRP5, CDCx)**: Performance improves as we scale up both the retriever backbone and the underlying language models (L407 and L427).
- **Efficiency (U5J6, CDCx)**: Compared to previous self-supervised methods, Revela uses approximately 1,000× less training data and 10× less compute (L370).

### **Responses to Reviewers**

In our rebuttal, we mainly clarified and strengthened the following points:

- **Reviewer ojFX, PRP5**: We clarified the **convenience and practicality of batch construction** in Rvela, where documents are simply split into chunks and directly assembled into batches, which can contain different topics.
- **Reviewer U5J6**: We further clarified Revela’s core motivation, casting retrieval as modeling inter-document dependencies, and detailed the algorithmic design, including how similarity scores parameterize in-batch attention, how local and cross-document contexts are combined, and how the retriever and LM are trained jointly.
- **Reviewer CDCx**: We highlighted Revela’s **stronger performance** compared to Replug and discussed potential reasons for this performance gap.

We are glad that all reviewers expressed strong and consistent recognition of Revela **before** the incident (AOE time):

- Reviewer **ojFX**: 8  (pending)
- Reviewer **U5J6**: 6 → 8 (25 Nov)
- Reviewer **PRP5**: 6 → 8 (23 Nov)
- Reviewer **CDCx**: 6 → 8 (26 Nov)

We thank the reviewers for their thoughtful feedback and constructive comments. And, we sincerely appreciate AC’s effort in this unusual discussion period and hope that our contributions and responses can be taken into full consideration.

Best regards,

Authors of Revela

---

### Meta-Review · Area_Chair_Lcc5 · 2026-01-05

**Summary:**

The paper proposes a new self-supervised training method, Revela, for training dense retrievers based on performing in-batch attention on retrieved text chunks which are then used for next-token prediction in language models. The in-batch attention is formulated in the paper, including a similarity metric that is used to condition the in-batch attention for similarity of chunks across documents which is one of the main challenges Revela aims to address. The paper applies Revela to three different evaluation benchmarks, including code (CoIR), reasoning (BRIGHT) and general domain (BEIR). The results generally show that Revela matches or exceeds the performance of state-of-the-art models on the evaluation benchmarks while providing distinct advantages in being self-supervised and requiring lower compute to train. The paper also provides some additional analysis on the properties of Revela which can inform future research directions.

The reviewers mainly noted the self-supervised nature and computational efficiency of the method as key strengths. Additional strengths highlighted by the reviewers include the ability of the method to work with diverse language modeling and retriever backbones without requiring additional labels. Some reviewers also praise the design of the experiments and ablations.

Most of the concerns of the reviewers related to clarifications related to the methodological choices and empirical settings that appear to have been clarified in the rebuttal. The authors also provided some additional results to addresses reviewer concerns during the rebuttal.

**Reviewer Concerns:**

Addressed Concerns:
* Reviewer ojFX's concerns seem generally addressed.
* Reviewer U5J6's concerns seem mostly addressed based on the author-reviewer discussion.
* Reviewer PRP5's concerns seem mostly addressed based on the author-reviewer discussion.
* Reviewer CDCx's concerns seem mostly addressed based on the author-reviewer discussion.

**Reviewer Scores:**

* Reviewer ojFX maintains score at 8.
* Reviewer U5J6 increases score to 8 based on the author-reviewer discussion.
* Reviewer PRP5 increases score to 8 based on the author-reviewer discussion.
* Reviewer CDCx increases score to 8 based on the author-reviewer discussion.

---

### Decision · Program_Chairs · 2026-01-26

Accept (Oral)